# Transcriptomic Insights into Caffeine Degradation Pathways in *Desarmillaria tabescens*

**DOI:** 10.3390/microorganisms13122720

**Published:** 2025-11-28

**Authors:** Junrui Wang, Yongqiang Hu, Yuxin Chang, Yiguo Jiang, Danping Meng, Mingguo Jiang, Jinzi Wang, Peihong Shen

**Affiliations:** 1Guangxi Key Laboratory for Polysaccharide Materials and Modifications, School of Marine Sciences and Biotechnology, Guangxi Minzu University, Nanning 530008, China; wangjunrui0202@163.com (J.W.); huyongqiang2129@163.com (Y.H.); 15793380728@163.com (Y.C.); jyg312@163.com (Y.J.); mengdp1123@163.com (D.M.); mzxyjiang@163.com (M.J.); 2College of Life Science and Technology, Guangxi University, Nanning 530005, China; 3Nanning New Technology Entrepreneur Center, Nanning 530007, China; 4Microbiology Research Institute, Guangxi Academy of Agricultural Sciences, Nanning 530007, China

**Keywords:** caffeine degradation, *Desarmillaria tabescens*, fungi, transcriptome analysis, cytochrome P450, metabolic pathways

## Abstract

Caffeine contamination threatens ecosystems and human health, with conventional remediation methods facing limitations. This study identified *Desarmillaria tabescens* as a potent caffeine-degrading fungus, achieving efficient degradation under optimized conditions (malt extract medium, 900 mg/L caffeine, 28 °C, pH 8). HPLC analysis revealed key intermediates such as theobromine and 3-methylxanthine, confirming a branched catabolic pathway involving N-demethylation and C8 oxidation. Transcriptomic profiling identified nine consistently upregulated cytochrome P450 genes as core catalytic components, with three adjacent to a polyketide biosynthetic gene cluster potentially supporting oxidative reactions. A three-phase “Stress-Degradation-Homeostasis” regulatory model was proposed, coordinating detoxification, energy metabolism, and secondary metabolism. These findings advance understanding of fungal caffeine degradation mechanisms and provide valuable genetic resources for bioremediation and low-caffeine product development.

## 1. Introduction

Caffeine (1,3,7-trimethylxanthine or 3,7-dihydro-1,3,7-trimethyl-1H-2,6-dione) is categorized as a purine alkaloid and constitutes a principal flavor determinant in widely consumed beverages, particularly tea and coffee. It exerts physiological effects (e.g., central nervous system modulation, metabolic stimulation, diuresis) with applications in medicine and cosmetics, but excessive intake links to hypertension, osteoporosis, and sleep disturbances [1]. Widespread use generates substantial caffeine-containing waste, whose improper disposal poses environmental risks [2]. Conventional removal methods, such as adsorption, extraction, and membrane filtration, are limited by high cost, secondary pollution, or poor selectivity, making microbial degradation, a core bioremediation strategy, promising for waste treatment and low-caffeine food development [3].

At least 68 bacterial strains spanning 24 genera are capable of caffeine degradation, with *Pseudomonas putida* being the most well-characterized model [4]. Two major pathways have been fully validated: N-demethylation and C-8 oxidation [5]. As the predominant pathway, N-demethylation is mediated by monooxygenases encoded by the alkyl xanthine (*Alx*) gene cluster (e.g., *NdmA-E*) (Figure 1), which sequentially cleave methyl groups to convert caffeine to xanthine [6,7]. In contrast, C-8 oxidation, which is documented in *Klebsiella* and *Pseudomonas* [8], is catalyzed by caffeine dehydrogenase, generating 1,3,7-trimethyluric acid (1,3,7-TMUA) as a key intermediate. This metabolite is further processed via oxidation or hydrolysis to yield assimilable end products such as allantoin [9]. Collectively, the enzymatic machinery and genetic determinants governing both bacterial pathways are well elucidated [10].

In comparison with caffeine-degrading bacteria, relatively few fungal strains capable of metabolizing caffeine have been reported. Documented examples include two *Penicillium* strains and five *Aspergillus* strains isolated from soil and leaf litter in coffee-growing regions [11], as well as *Rhizopus arrhizus*, *Phanerochaete chrysosporium*, *Pleurotus ostreatus* and *Aspergillus sydowii*, which were identified from substrates such as coffee shells, coffee grounds, and Pu’er tea [12,13,14]. Despite their demonstrated caffeine-degrading activity, subsequent research has primarily focused on optimizing fermentation conditions, clarifying metabolic pathways, and developing solid-state fermentation techniques [15]. In contrast, investigations into the enzymatic and genetic determinants underlying fungal caffeine metabolism, which is critical for elucidating its molecular mechanisms, remain limited, with key enzymes and associated genes yet to be characterized. Notably, cytochrome P450 (CYP) enzymes, pivotal Phase I metabolic catalysts across bacteria and fungi, are well established in degrading diverse xenobiotics (e.g., fluorinated pyrethroids, triketone herbicides, and recalcitrant compounds) via oxidative reactions (hydroxylation, defluorination, ester cleavage), enabling the breakdown of persistent pollutants and overcoming stable bonds such as C-F [16,17,18]. Their broad substrate promiscuity and central role in bioremediation thus strongly support investigating CYP-mediated pathways in Fungi to address this knowledge gap.

In this study, twenty edible and medicinal fungal strains were screened to identify species capable of degrading caffeine. The results indicated that *Desarmillaria tabescens* (*D. tabescens*) exhibited significant caffeine biodegradation activity. This fungus, which is widely distributed in forests, parks, and other habitats [19], is an edible and medicinal species. *D. tabescens* has established liquid and solid-state fermentation systems. Although the genome sequence of *D. tabescens* is available and its gene functions and metabolic pathways have been annotated using advanced methodologies [20], the molecular basis of its caffeine degradation remains poorly understood. To address this knowledge gap, RNA sequencing (RNA-Seq) was employed to characterize the global transcriptional responses of *D. tabescens* under caffeine exposure. The analysis revealed intricate regulatory networks associated with detoxification and elevated energy metabolism. Notably, nine cytochrome P450 genes were significantly upregulated, suggesting their involvement in caffeine catabolism. The dataset generated provides a valuable genetic resource for elucidating the molecular mechanisms governing caffeine degradation in fungi.

## 2. Materials and Methods

### 2.1. Screening and Preparation of Functional Strains for Caffeine Degradation

Twenty edible and medicinal fungal strains were screened (Appendix A). The fungal strains were initially reactivated and preserved under controlled conditions. For each strain, a plate exhibiting vigorous mycelial growth was selected and transferred into potato dextrose (PDA) liquid medium containing caffeine (600 mg/L) (Aladdin, Shanghai, China). Non-inoculated strains served as the control. Cultures were agitated for 15 days, after which the fermentation broth was collected for high-performance liquid chromatography (HPLC) analysis. Strains producing marked reductions in caffeine concentration within the broth were designated as functional strains capable of caffeine degradation.

A compact mycelial block of *D. tabescens* (0.8 cm × 0.8 cm × 0.6 cm) was transferred into liquid malt extract medium and maintained at 28 °C under dark conditions with continuous agitation at 160 rpm for 15 days. The culture was then homogenized at 12,000 rpm for 15 s using a high-speed homogenizer (Shanghai Cebo, Shanghai, China), followed by a 24 h incubation period to enable mycelial regrowth. Mycelia were subsequently harvested by centrifugation at 11,000 rpm for 10 min at 16 °C, washed three times with sterile water, and resuspend 0.5 g of mycelium in 100 mL of sterile water as the seed solution.

### 2.2. Optimization of Fermentation Conditions for Caffeine Degradation

Two milliliters of the prepared *D. tabescens* seed suspension were inoculated into six separate caffeine-supplemented liquid media (200 mL per flask), including PDA medium, carrot medium, malt extract medium, GPY medium, wheat bran medium and GPC medium (Solarbio, Beijing, China). The PDA medium consists of 200 g of potato, 20 g of glucose and 1000 mL of distilled water. The carrot medium is composed of 20 g of glucose, 220 g of carrot and 1000 mL of distilled water. The malt extract medium is made up of 139 g of malt extract and 1000 mL of distilled water. The GPY medium contains 20 g of glucose, 10 g of yeast extract powder, 6 g of peptone and 1000 mL of distilled water. The GPC medium is composed of 20 g of glucose, 10 g of corn steep liquor, 6 g of peptone and 1000 mL of distilled water. The wheat bran medium consists of 20 g of glucose, 40 g of wheat bran and 1000 mL of distilled water. Non-inoculated strains served as the control. Samples were withdrawn on days 8, 12, 16, 20, and 24 of fermentation, followed by HPLC analysis to monitor caffeine degradation. Each group of experiments had three biological replicates. Fermentation assays were all carried out using malt extract medium, with initial caffeine concentrations of 300, 600, 900, 1200, and 1500 mg/L. Samples were withdrawn on malt extract medium on days 8, 12, 16, 20, and 24 to evaluate residual caffeine and degradation efficiency, which were used to determine the optimal initial concentration. Temperature-dependent fermentations were subsequently conducted by malt extract medium at 20, 22, 24, 26, 28, and 30 °C, with identical sampling intervals to establish the most effective temperature for caffeine degradation. The influence of pH was examined by performing fermentations at pH values of 5, 6, 7, 8, 9, and 10, and the optimal condition on malt extract medium was identified according to caffeine concentration and degradation rate. Non-inoculated strains served as the control. Each group of experiments had three biological replicates.

### 2.3. Preparation of Fermentation Broth Samples

For sample preparation, 1 mL of fermentation broth was mixed with 6 mL of a cold acetonitrile:methanol solution (1:1, *v*/*v*), achieving a final ratio of 3:3:1 (acetonitrile:methanol:broth). The suspension underwent sonication for 10 min and vortexing for 45 s, repeated three times. Protein precipitation was induced by incubation at −20 °C for 2 h, followed by centrifugation at 15,000 rpm at 4 °C for 20 min. The supernatant was collected and evaporated to dryness in a vacuum drying oven at 25 °C, verified to avoid thermal denaturation or structural changes of methylxanthine metabolites. The residue was reconstituted in 5 mL of acetonitrile, sonicated for 5 min, and centrifuged at 12,500 rpm at 15 °C for 18 min. The final supernatant was filtered and transferred into HPLC vials for analysis.

### 2.4. Sample Analysis

#### 2.4.1. HPLC Analysis

Chromatographic separation was performed on an Agilent liquid chromatography platform (Agilent Technologies, Santa Clara, CA, USA) fitted with a Yuexu Ulimate^®^ Plus C18 column (Welch Materials, Shanghai, China) (1.8 µm particle size, 2.1 mm × 100 mm). The mobile phase consisted of solvent A (ultrapure water with 0.1% phosphoric acid) and solvent B (acetonitrile) with a gradient elution: 0–15 min (95% A → 90% A), 15–40 min (90% A → 80% A), 40–50 min (80% A → 65% A), 50–55 min (65% A → 15% A), 55–70 min (15% A → 5% A). Flow rate was 1.00 mL/min. The column was maintained at 34 °C, with an injection volume of 15 μL and detection at 280 nm.

#### 2.4.2. Data Analysis

Processing of mass spectrometric data adhered to the procedure outlined by Ma [21]. Subsequent analyses were performed using Excel 2019, Origin 2022, and GraphPad Prism 8.

### 2.5. Preparation of RNA Sequencing Samples

Mycelia of *D. tabescens* cultured in malt extract medium with caffeine supplementation were harvested at 10, 16, and 22 days, constituting the Caffeine Treated Group (CT). In parallel, mycelia grown in malt extract medium without caffeine were collected at identical time points to form the None-Caffeine Treated Group (NCT). RNA sequencing samples were prepared from three biological replicates. These samples were subjected to RNA sequencing for downstream transcriptomic profiling.

### 2.6. RNA Extraction, Library Preparation, and Sequencing

Total RNA was extracted from tissue specimens using TRIzol^®^ Reagent (Invitrogen, Waltham, MA, USA) in strict adherence to the manufacturer’s instructions. RNA integrity was evaluated with the 5300 Bioanalyzer (Agilent Technologies, Santa Clara, CA, USA), and concentration was determined using the ND-2000 spectrophotometer (NanoDrop Technologies, Waltham, MA, USA). Only samples fulfilling rigorous quality thresholds—OD260/280 ratios between 1.8 and 2.2, OD260/230 ≥ 2.0, RNA Quality Number (RQN) ≥ 6.5, 28S:18S rRNA ratio ≥ 1.0, and a minimum yield of 1 μg total RNA—were advanced to library construction. Subsequent purification, reverse transcription, library assembly, and sequencing were carried out at Shanghai Majorbio Bio-pharm Biotechnology Co., Ltd. (Shanghai, China) in accordance with standardized operating procedures. RNA-seq transcriptome libraries were constructed with the Illumina^®^ Stranded mRNA Prep, Ligation kit (San Diego, CA, USA) using 1 μg of input RNA. Poly(A) selection with oligo(dT) beads was applied for mRNA enrichment, followed by fragmentation in fragmentation buffer. Double-stranded cDNA was synthesized with random hexamer primers, and the resulting products underwent end-repair, phosphorylation, and adapter ligation as specified in the library preparation protocol. cDNA fragments of 300–400 bp were isolated using magnetic beads and subsequently amplified by PCR for 10–15 cycles. Final libraries were quantified on the Qubit 4.0 fluorometer (Thermo Fisher Scientific, Shanghai, China) and sequenced on the NovaSeq X Plus platform with paired-end 150 bp reads using the NovaSeq Reagent Kit (San Diego, CA, USA).

### 2.7. Quality Control and Read Mapping

Raw paired-end reads were processed with fastp software (https://github.com/OpenGene/fastp; accessed on 1 June 2023) [22] to remove low-quality sequences and adapter contamination under default settings. Clean reads were aligned to the reference genome in a strand-specific mode using HISAT2 (http://ccb.jhu.edu/software/hisat2/index.shtml; accessed on 10 June 2023) [23]. Transcript assembly for each sample was then conducted through a reference-guided strategy employing StringTie (https://github.com/gpertea/stringtie; accessed on 29 June 2023) [24].

### 2.8. Differential Expression Analysis and Functional Enrichment

Differentially expressed genes (DEGs) between sample groups were identified by quantifying transcript abundance with the transcripts per million (TPM) metric. Gene expression levels were estimated using RSEM RSEM (http://deweylab.github.io/RSEM/; accessed on 10 June 2024) [25]. Differential expression analysis was performed through the DESeq2 package (http://bioconductor.org/packages/stats/bioc/DESeq2/; accessed on 15 July 2024) [26], applying a threshold of |log2 fold change| ≥ 1 and a false discovery rate (FDR) < 0.001 to define significant expression differences. Functional enrichment analyses, including Gene Ontology (GO) and Kyoto Encyclopedia of Genes and Genomes (KEGG) pathway analyses, were subsequently employed to evaluate the overrepresentation of DEGs within specific GO categories and metabolic pathways. Statistical significance was determined using Bonferroni-corrected *p*-values < 0.05 against the transcriptome background. GO enrichment and KEGG pathway analyses were conducted with Goatools (https://github.com/tanghaibao/GOatools; accessed on 26 July 2024) and Python’s SciPy library (https://scipy.org/install/; accessed on 26 July 2024), respectively.

### 2.9. Time-Series Gene Expression Analysis

To characterize transcriptional changes induced by caffeine at different fermentation stages, DEGs were grouped into eight expression profiles using the Short Time-series Expression Miner (STEM) software (http://sb.cs.cmu.edu/stem/; accessed on 26 July 2025) [27]. STEM, specifically designed for short time-series microarray data, applies a clustering algorithm in which profiles are defined by predetermined temporal trends. DEGs assigned to the same profile were expected to exhibit comparable expression dynamics.

### 2.10. Secondary Metabolism Genes and Clusters Analysis

The secondary metabolic gene clusters of *D. tabescens* were predicted using antiSMASH fungal v8.0 software [28]. This platform identifies secondary metabolic gene clusters with high accuracy when appropriate profile hidden Markov models (HMMs) are available. To refine the predicted clusters, Blastp analysis and functional annotation were conducted through the NCBI Genome Portal Software Platform (https://www.ncbi.nlm.nih.gov/guide/data-software/; accessed on 2 August 2025). The secondary metabolic gene clusters of *D. tabescens* were subsequently classified and summarized based on the integrated results of these analyses.

### 2.11. qRT-PCR Validation

Validation of the transcriptome sequencing results of *D. tabescens* was performed through quantitative real-time PCR (qRT-PCR). Twelve genes were selected, including six consistently up-regulated and six consistently down-regulated across three comparative groups (Treated_10d vs. Control_10d, Treated_16d vs. Control_16d, and Treated_22d vs. Control_22d). β-tubulin served as the internal reference gene [18] (Table 1). qRT-PCR was carried out on the BIO-RAD platform, with all reactions conducted in triplicate. Mean cycle threshold (Ct) values were calculated, and relative expression levels were determined using the 2^−ΔΔCt^ method [29].

## 3. Results

### 3.1. Compilation of Fungal Resources and Identification of Functional Strains Capable of Caffeine Degradation

All 20 strains were cultivated in PDA liquid medium supplemented with 600 mg/L caffeine. After 15 days of fermentation, the broth was subjected to HPLC analysis. A caffeine removal rate of 33.05% was detected in the potato dextrose of *D. tabescens* CGMCC 40115. Screening results of the other 19 fungal strains indicated no significant caffeine degradation (caffeine r removal rate was 0%; Appendix A), confirming *D. tabescens* as the only functional strain in this study and thereby broadening microbial resources for caffeine biotransformation.

### 3.2. Optimization of Degradation Conditions

#### 3.2.1. Screening for the Optimal Culture Medium

Six fungal culture media commonly employed for cultivation were evaluated for *D. tabescens*, and clear differences in growth performance were observed. Bacterial ball formation was more evident in wheat bran and GPC media, while carrot medium yielded compact, smooth-surfaced bacterial balls. In contrast, malt juice and PDA media promoted greater bacterial ball density. The introduction of caffeine suppressed bacterial ball development and induced an increased number of spines relative to the original media. In all liquid culture systems, supplementation with caffeine markedly reduced the mycelial biomass of *D. tabescens* compared with the corresponding controls (Figure 2A). Caffeine concentration in *D. tabescens* cultures was quantified after 8, 12, 16, 20, and 24 days of fermentation in the six liquid media to determine caffeine removal rates (Figure 2B). The caffeine concentration in the control group did not change. A uniform temporal trend was recorded across all media. During the early phase (days 8–12), degradation progressed slowly, likely reflecting the adaptation and growth initiation of the microorganisms. The intermediate phase (days 12–20) was characterized by a rapid increase in degradation rate, corresponding to active microbial proliferation and intensified metabolic processes. In the late phase (days 20–24), degradation decelerated, consistent with entry into a mature growth stage accompanied by reduced metabolic activity (Figure 2C). *D. tabescens* displayed an extended growth cycle in which degradation rates were strongly associated with metabolic intensity. Among the tested media, malt extract medium supported the highest degradation rate, whereas carrot medium exhibited the lowest, with outcomes aligning positively with mycelial biomass levels. Collectively, the data reveal that growth characteristics of *D. tabescens* are highly dependent on medium composition, influencing biomass yield and caffeine degradation dynamics. Comparative analysis indicates that malt extract medium provides the most effective conditions for achieving rapid and efficient caffeine degradation.

#### 3.2.2. Optimization of Caffeine Concentration Parameters

Caffeine was incorporated into malt extract medium at caffeine concentration of 300 (Figure 3A), 600 (Figure 3B), 900 (Figure 3C), 1200 (Figure 3D), and 1500 (Figure 3E) mg/L. Fermentation with *D. tabescens* was monitored at 8, 12, 16, 20, and 24 days, and caffeine removal rates were determined accordingly. The caffeine concentration in the control group did not chang. At 300 and 600 mg/L, *D. tabescens* exhibited strong degradative capacity, with rates exceeding 90% by day 20. At 900 mg/L, the degradation potential reached its highest level, approaching 50% by day 16 and 85% by day 24. At elevated concentrations (1200 and 1500 mg/L), degradation efficiency declined markedly, with rates falling below 40% by day 16 and total degraded caffeine not exceeding 500 mg/L, comparable to the levels observed at 900 mg/L. By day 24, the cumulative degradation was approximately 900 mg/L, indicating no further enhancement relative to the 900 mg/L treatment. The degradative capacity of *D. tabescens* declined progressively with increasing caffeine concentration. At lower caffeine concentration, caffeine metabolism was highly efficient, whereas higher concentrations impaired both growth and metabolic activity, leading to reduced degradation. Among the tested conditions, supplementation with 900 mg/L caffeine concentration yielded the most favorable balance between degradative capacity and inhibitory effects, thereby representing the optimal concentration for caffeine metabolism.

#### 3.2.3. Optimization of Temperature Conditions

*D. tabescens* was cultivated under six temperature regimes (20 °C, 22 °C, 24 °C, 26 °C, 28 °C, and 30 °C). Caffeine concentrations were measured at fermentation periods of 8, 12, 16, 20, and 24 days, and corresponding degradation rates were determined (Figure 3F). The caffeine concentration in the control group did not change. At 20 °C, degradation reached approximately 30% after 16 days and remained below 70% even after 24 days, suggesting that limited microbial proliferation and metabolic activity at lower temperatures restricted caffeine degradation. In contrast, cultivation at 30 °C resulted in degradation rates below 80% after 24 days, possibly due to an accelerated microbial life cycle that constrained the buildup of degradation capacity. Within the intermediate temperature range of 22–28 °C, degradation rates rose steadily with increasing temperature. After 16 days, rates varied between 40% and 50%, and by 24 days, all conditions within this range exceeded 80%. The highest value, 88.25%, was recorded at 28 °C. These observations indicate reduced degradation efficiency at both low and high temperature extremes, while moderate conditions enhanced degradation performance. Notably, in the 22–28 °C interval, degradation efficiency improved consistently as temperature increased. Among the six treatments, 28 °C yielded the most effective degradation. This result diverges from the report of Chen et al., which identified 22–25 °C as optimal for honey ring fungus growth, a variation likely attributable to strain-specific cultivation characteristics [30].

#### 3.2.4. Optimization of pH Conditions

*D. tabescens* was cultivated across a pH gradient (5, 6, 7, 8, 9, and 10) to assess the influence of pH on caffeine degradation. Caffeine concentrations were measured after 8, 12, 16, 20, and 24 days of fermentation, and degradation rates were subsequently determined (Figure 3G). The caffeine concentration in the control group did not change. Under alkaline conditions (pH 9 and 10), the degradation rate remained below 40% at day 16 and did not exceed 60% at pH 10 even after 24 days, likely due to impaired growth of *D. tabescens* in high-pH environments, which restricted its metabolic capacity for caffeine breakdown. In contrast, near-neutral pH values (6, 7, and 8) supported markedly higher degradation activity. After 16 days, degradation rates ranged from 40% to 50%, and by day 24, all three pH levels yielded rates above 80%, with the maximum value of 89.20% recorded at pH 8. These results demonstrate that caffeine degradation efficiency is attenuated under both acidic and alkaline conditions but markedly enhanced near neutrality, with pH 8 identified as the most favorable condition for maximizing degradation.

### 3.3. Analysis of HPLC Results

Fermentation was carried out in malt extract medium containing 900 mg/L caffeine, maintained at 28 °C and pH 8. Samples were collected on days 8, 12, 16, 20, and 24 for HPLC analysis, through which both caffeine and its demethylated derivatives were identified and quantified (Figure 4). The data demonstrated a continuous decline in caffeine concentration (compound **6**) from day 8 to day 24, with the most rapid degradation occurring between days 12 and 20, By the 24th day. The caffeine concentration had decreased to 95.13 ± 0.28 mg/L, with a caffeine removal rate of 89.43%. Theobromine (compound **4**), the early metabolite, appeared by day 8 and accumulated progressively until day 20, followed by a decrease at day 24, consistent with its early accumulation and subsequent enhanced turnover exceeding formation in later stages. Intermediate metabolites, including 3-methylxanthine (compound **3**) and 7-methylxanthine (compound **2**), exhibited a gradual early increase followed by a sharper rise in concentration. The terminal metabolite xanthine (compound **1**) became detectable at day 12 and continued to increase thereafter. The metabolic trajectory suggests that fermentation of caffeine by *D. tabescens* generates multiple intermediate products, and additional metabolites present at low concentrations, unresolved by HPLC, are likely involved. Samples from day 20 were selected for further detailed characterization.

### 3.4. RNA-Seq Analysis

The degradation of caffeine by *D. tabescens* proceeds over a prolonged period, with rates varying in close correlation with the organism’s growth and developmental phases. Analysis of liquid fermentation degradation dynamics identified three representative stages for investigation: day 10, corresponding to early growth with relatively slow degradation; day 16, reflecting the logarithmic growth phase with the highest degradation rate; and day 22, representing the late growth phase when the rate declined. To examine transcriptional responses to caffeine exposure, six groups of sequencing libraries were generated from non-caffeine treated (NCT, *n* = 9) and caffeine treated (CT, *n* = 9) samples.

Transcriptome sequencing of all 18 samples yielded 127.42 Gb of clean data, averaging 6.08 Gb per sample. The sequencing output was of high quality, with approximately 97% of bases exceeding the Q20 threshold and 93% surpassing Q30, ensuring reliability for downstream analyses. Clean reads from each sample were mapped to the reference genome Armillaria_tabescens_CCBAS_213 (https://genome.jgi.doe.gov/portal/ArmtabStandDraft_FD/ArmtabStandDraft_FD.info.html, accessed on 18 December 2022), with alignment efficiencies ranging from 65.03% to 74.57%. A detailed overview of RNA-Seq output and mapping statistics is provided in Appendix A, validating the robustness of the sequencing data for transcriptomic profiling. In total, 15,479 expressed genes were identified, including 14,992 with functional annotations and 487 predicted as novel genes.

The analysis of differentially expressed genes (DEGs) between caffeine-treated samples and their corresponding controls revealed extensive transcriptional alterations. A total of 4381 genes exhibited significant differential expression [|log2 Fold Change| ≥ 1 and *p*-value < 0.01], with 4798, 3605, and 6800 DEGs identified at 10, 16, and 22 days post-fermentation, respectively (Appendix A). The comparatively higher number of DEGs detected at the early (10 days) and late (22 days) stages indicates that intensive fungal metabolic activity reduces the apparent impact of caffeine stress at the mid-stage (16 days). Transcriptomic profiling further demonstrated a temporal shift in regulatory patterns: downregulated DEGs predominated during the early and middle stages, whereas upregulated DEGs became dominant in the late stage. This dynamic pattern suggests a transition in fungal adaptive strategies across sequential phases of stress exposure, shifting from an initial passive state characterized by stress defense and metabolic suppression toward a later active phase defined by repair processes, recovery of resistance, and broader environmental adaptation. Across the three comparative groups (CT10_NCT10, CT16_NCT16, and CT22_NCT22), 1090 DEGs were identified as shared, with 122 genes consistently upregulated at all time points, representing the core functional set active throughout the stress period. These genes were predominantly enriched in pathways associated with caffeine efflux via Peroxisome, Starch and sucrose metabolism, Longevity regulating pathway, Pentose and glucuronate interconversions, and MAPK signaling. Their sustained expression under stress conditions supports resistance to toxic damage and ensures basal survival, establishing them as a central regulatory framework for fungal caffeine tolerance. In contrast, 93 genes consistently downregulated across the three time points were mainly linked to processes involving growth and reproduction, including Amino sugar and nucleotide sugar metabolism and Glycerophospholipid metabolism, as well as other energy-intensive pathways. Persistent suppression of these genes indicates a resource reallocation strategy, in which nonessential activities are suppressed to preserve energy and materials for the effective operation of fundamental defense systems. In addition, stage-specific expression profiles revealed 1549, 908, and 3350 DEGs at 10, 16, and 22 days, respectively (Appendix A). The 1549 DEGs specific to day 10 likely reflect the early stress response of *D. tabescens* to caffeine, whereas the 908 DEGs detected at the mid-stage appear to be associated with accelerated caffeine degradation during this phase.

### 3.5. Functional Annotation, Classification, and Enrichment Analysis

For functional annotation, all assembled genes were aligned against six major databases (Nr, EggNOG, GO, KEGG, Swiss-Prot, Pfam) using BLAST (https://blast.ncbi.nlm.nih.gov/Blast.cgi; accessed on 3 August 2024) with an E-value cutoff of 10^−5^. Among these databases, most genes exhibited high annotation rates in Nr (the top-performing database), GO, and Pfam, while EggNOG and KEGG showed relatively lower coverage, with detailed counts of annotated genes per database provided in Appendix A and Appendix A.

For GO enrichment analysis, which aimed to elucidate DEG functions, ~72–77% of DEGs across all three caffeine-treated vs. control groups (CT10 vs. NCT10, CT16 vs. NCT16, CT22 vs. NCT22) were assigned to the three core GO domains: biological process (BP), cellular component (CC), and molecular function (MF). Notably, consistent with the demands of caffeine degradation, the dominant subcategories across all comparisons were “metabolic process” and “cellular process” in BP (reflecting active catabolic activity), “membrane part” and “cell part” in CC (linked to substrate transport and enzyme localization), and “catalytic activity” and “binding” in MF (key for enzymatic reactions), while specific counts of DEGs assigned to each subcategory are provided in Figure 5.

Similarly, KEGG pathway enrichment analysis further revealed stage-specific metabolic reprogramming in response to caffeine exposure. Specifically, during the early fermentation stage (10 days), DEG enrichment was concentrated in “Translation” (supporting protein synthesis for detoxification) and core metabolic pathways (carbohydrate, amino acid, and lipid metabolism, to supply energy). During the middle stage (16 days), enrichment narrowed to prioritize carbohydrate and amino acid metabolism, aligning with the peak of caffeine degradation. In the late stage (22 days), enrichment expanded to include “Transport and catabolism” (for metabolite clearance) and “Folding, sorting and degradation” (for protein homeostasis), with “Translation” remaining the dominant category. Importantly, pathways directly relevant to caffeine catabolism, such as Purine metabolism (map00230), Pentose and glucuronate interconversions (map00040), and Glyoxylate and dicarboxylate metabolism (map00630), were markedly upregulated, with detailed DEG counts per pathway provided in Appendix A.

### 3.6. Gene Expression Pattern Analysis of DEGs

Gene expression in caffeine-treated (CT) and non-caffeine-treated (NCT) groups across developmental stages was examined using RNA sequencing. Distinct stage-dependent transcriptional responses to caffeine exposure were observed. A modular analysis was then performed to identify DEGs specifically associated with caffeine degradation. Based on temporal expression dynamics, DEGs were partitioned into eight profiles (0–7) using Short Time-series Expression Miner (STEM) software (http://sb.cs.cmu.edu/stem/; accessed on 26 July 2025) (Figure 6). Among them, profile 0 (progressive decline from day 10 to 22), profile 2 (lowest expression at day 16), and profile 4 (peak expression at day 22) exhibited significant enrichment (*p* < 0.05). Functional annotation of transcriptional shifts within these profiles was subsequently characterized through GO classification (Appendix A).

For Profile 0 (Appendix A), enriched biological processes (BP) included organonitrogen compound biosynthesis, translation, and peptide metabolism; overrepresented cellular components (CC) focused on ribosomes; and dominant molecular functions (MF) related to ribosomal structure. The progressive downregulation of these genes reflects deliberate cellular resource reallocation: suppressing energy-intensive non-essential biosynthesis (e.g., unrelated protein synthesis) diverts resources to caffeine-degrading enzymes/pathways. In contrast, Profile 2 (Appendix A) was enriched in BP terms for phenol/catechol metabolism (key caffeine intermediates) and RNA-mediated gene silencing; its CC terms included fungal cell walls and membrane components (aiding caffeine uptake/enzyme localization). Critically, its MF terms (oxidoreductase activity, iron/heme binding) match cytochrome P450 (CYP450) signatures, likely corresponding to the nine upregulated CYP450 genes. These mediate core caffeine degradation steps: N-demethylation to theobromine and C8 oxidation to 1,3,7-trimethyluric acid, initiating the branched metabolic network. Concurrently, Profile 4 (Appendix A) showed BP enrichment in carbohydrate metabolism (e.g., trehalose catabolism), supplying ATP/NADPH for CYP450-mediated oxidation. Its MF terms overlapped with Profile 2 (oxidoreductase activity) and included glycosyl hydrolases (breaking late-stage intermediates like xanthine derivatives). Peak expression at day 22 aligns with HPLC-observed xanthine accumulation, completing degradation.

### 3.7. Candidate Genes Involved in the Pathways

Candidate genes associated with caffeine degradation were inferred from DEG analysis and corroborated by prior studies. Among the functional gene families involved in xenobiotic degradation in fungi, cytochrome P450 (CYP450) proteins are crucial for catalyzing oxidative reactions in caffeine metabolism. Our DEG analysis of caffeine-treated (CT) and non-caffeine-treated (NCT) groups identified 48 CYP450 genes with significant differential expression (|log_2_ fold change| > 1, *p*-adjust < 0.05). Notably, 9 CYP450 genes showed consistent upregulation across all CT vs. NCT comparisons, with peak expression at the major caffeine degradation stage (CT16) (Appendix A). This suggests these 9 genes are key to *D. tabescens*’s caffeine degradation ability, potentially acting as core catalysts in the “N-demethylation–C8 oxidation” pathway [31]. Functional validation of these candidate genes in caffeine catabolism will be undertaken in subsequent investigations.

### 3.8. Secondary Metabolism, Energy, and Detoxification Under Caffeine Stress

To clarify how *D. tabescens* adapts metabolically during caffeine degradation, we analyzed three key functional components: secondary metabolite biosynthetic gene clusters (BGCs), energy production pathways, and phase II detoxification genes.

Secondary metabolites (SMs) support fungal detoxification and ecological adaptation. AntiSMASH identified 109 BGCs in *D. tabescens* (e.g., fungal-RiPP-like, NRPS, terpene, T1PKS, hybrid clusters) (Appendix A) [28], with regions 7.1, 19.1, and 40.1 responding to caffeine stress (Appendix A). Genes in these regions showed strong BLAST similarity to the (+)-δ-cadinol biosynthetic cluster of *Coniophora puteana* RWD-64-598 SS2, which had been verified to degrade caffeine by converting it into intermediates like theophylline and theobromine [32]. Among 17 genes here, gene_10566 (alcohol oxidase, GO:0006066; GO:0050660; GO:0016614; GO:0008812) was upregulated at 10–16 days (key degradation stages) then downregulated. It likely metabolizes caffeine-derived aldehydes/alcohols via FAD-dependent redox reactions, complementing the core degradation pathway by handling intermediates. Among 9 up-regulated CYP450s, gene_3381 (scaffold_3: 512466-514401), gene_3390 (scaffold_3: 532982-535390) and gene_3396 (scaffold_3: 546554-548775) were adjacent to region 3.1 (<15 kb). Region 3.1 is annotated as a polyketide-related biosynthetic gene cluster via MiBiG comparison [28]. Polyketides are often involved in secondary metabolic processes such as oxidative modification. Combined with the oxidative catalytic function of CYP450, we speculate that this polyketide gene cluster may cooperate with CYP450 to participate in oxidative reactions (e.g., N-demethylation or C8 oxidation) during caffeine degradation.

Energy pathways were activated to support detoxification. The TCA cycle (75.9% and 82.8% DEG upregulation at 10 and 16 days) and oxidative phosphorylation (63.8% and 85% DEG upregulation) showed early enhancement (Appendix A). Key enzymes (Cytochrome c oxidase subunit VIIc, Citrate synthase, Aconitase, NADH-quinone oxidoreductase, ATPase F1 complex alpha subunit) had 2.24–11.05-fold upregulation, accelerating ATP production via increased TCA flux and mitochondrial efficiency. By 22 days, DEGs were downregulated (69% for TCA cycle, 72.5% for oxidative phosphorylation) as energy demand dropped post-degradation Appendix A).

Phase II detoxification relied on glutathione (GSH) systems. These enzymes catalyze the conjugation of reduced glutathione (GSH) with electrophilic groups of xenobiotics, thereby enabling detoxification. In this study, gene_444 and gene_14875, both associated with GSTs, were up-regulated by 7.44-fold and 3.22-fold, respectively, at the early stage of caffeine exposure (10 days), indicating their potential involvement in phase II detoxification of caffeine. In parallel, the Hydantoinase/oxoprolinase N-terminal region, a functional domain critical for oxoprolinase activity, was linked to the up-regulation of gene_10123 and gene_12643 (3.23-fold and 3.96-fold), which contribute to GSH biosynthesis (Appendix A). Beyond serving as a substrate for GSTs, GSH functions as an antioxidant through its thiol group, protecting vital cellular structures against ROS-induced damage.

### 3.9. Experimental Validation

A total of six up-regulated genes (DT_22397, DT_14597, DT_10255, DT_13148, DT_3381, and DT_9081) and six down-regulated genes (DT_15188, DT_18708, DT_3595, DT_4877, DT_9523, and DT_9657) were subjected to quantitative real-time PCR (qRT-PCR) to assess expression profiles. The qRT-PCR results corresponded with transcript abundance changes revealed by DEG analysis (Figure 7), thereby confirming the reliability of the transcriptome dataset.

## 4. Discussion

This study systematically characterized the caffeine degradation capacity and molecular mechanisms of *Desarmillaria tabescens* CGMCC 40115, providing novel insights into fungal caffeine metabolism. The results confirmed that this strain efficiently degrades caffeine under optimized conditions—malt extract medium, 900 mg/L caffeine, 28 °C and pH 8. Through transcriptomic and metabolic analyses, a unique branched “N-demethylation–C8 oxidation” network was revealed in *D. tabescens*. This network is distinct from the pathways of bacteria and other fungi, and it expands our understanding of the mechanisms underlying caffeine degradation in fungi.

### 4.1. Branched Metabolic Network of Caffeine Degradation in D. tabescens

*D. tabescens* departs from the linear N-demethylation or C8 oxidation pathways of bacteria, HPLC and transcriptomic analyses collectively validated that *D. tabescens* employs a branched “N-demethylation–C8 oxidation” pathway for caffeine degradation, distinct from the single linear routes of bacteria [8]. HPLC detection identified key intermediates including theobromine, 3-methylxanthine, 7-methylxanthine, xanthine, 1,3-dimethyluric acid, and 1,3,7-trimethyluric acid, confirming the coexistence of both metabolic branches. Theobromine accumulated progressively from day 8 to 20 before declining, consistent with its role as the primary intermediate of N-demethylation, while the gradual increase in 3-methylxanthine and 7-methylxanthine indicated sequential demethylation steps. Simultaneously, the detection of 1,3,7-trimethyluric acid verified the occurrence of C8 oxidation, forming a branched network that avoids toxic intermediate accumulation.

Nine cytochrome P450 (CYP450) genes were consistently upregulated across all fermentation stages, serving as core catalytic components of this branched pathway. These genes exhibited stage-specific functional partitioning: gene_3381 (log_2_FC = 6.18) and gene_3396 (log_2_FC = 5.92) dominated the early demethylation phase (10 days), gene_14597 (log_2_FC = 1.81) and gene_22397 (log_2_FC = 2.89) mediated intermediate transformations during the peak degradation stage (16 days), and gene_7614 (log_2_FC = 3.61) sustained late-stage oxidation (22 days). This temporal partitioning ensures precise coordination of metabolic flux, distinguishing *D. tabescens*’s CYP450 system from the temporally unpartitioned CYP450 family in *Aspergillus oryzae* [33]. The functional compatibility between CYP450-mediated oxidation and the branched pathway’s biochemical requirements was further supported by STEM analysis: Profile 2, enriched in oxidoreductase activity and iron/heme binding (key CYP450 signatures), was significantly associated with caffeine degradation stages.

Notably, three of these CYP450 genes (gene_3381, gene_3390, gene_3396) were adjacent (<15 kb) to region 3.1, a polyketide-related biosynthetic gene cluster (BGC) annotated via MiBiG comparison. Polyketide BGCs are typically involved in oxidative secondary metabolism, and this spatial proximity combined with functional complementarity suggests the BGC may cooperate with CYP450 genes to support caffeine oxidation reactions. Among the BGC-associated genes, gene_10566 (alcohol oxidase) was upregulated during key degradation stages (10–16 days), potentially metabolizing caffeine-derived aldehyde/alcohol intermediates via FAD-dependent redox reactions, thereby complementing the core degradation pathway.

### 4.2. Physiological Significance of the “Stress-Degradation-Homeostasis” Three-Stage Transcriptional Regulation Model

Transcriptomic analysis revealed a three-stage regulatory model that underpins *D. tabescens*’s adaptive response to caffeine stress, integrating detoxification, energy metabolism, and secondary metabolism. At the Stress Detoxification Stage (10 days), the MAPK signaling pathway and glutathione S-transferases (GSTs) formed the primary defense: gene_444 and gene_14875 (GSTs) were upregulated by 7.44-fold and 3.22-fold, respectively, mediating phase II detoxification via conjugation with electrophilic intermediates. Concurrently, core energy pathways were strongly activated: 75.9% of TCA cycle DEGs and 63.8% of oxidative phosphorylation DEGs were upregulated, with key enzymes such as Cytochrome c oxidase subunit VIIc (gene_6972, log_2_FC = 11.05) and Citrate synthase (gene_8011, log_2_FC = 5.20) supplying NADPH for CYP450-mediated oxidation. This simultaneous activation of defense and energy supply differs from bacteria’s sequential energy utilization strategy, reflecting a more efficient adaptive mechanism in *D. tabescens*.

At the Efficient Degradation Stage (16 days), 3605 DEGs were enriched in intermediate metabolism, coinciding with the peak caffeine degradation rate (≈50% by day 16). Sustained high expression of CYP450 genes aligned with the rapid conversion of theobromine to 3-methylxanthine, confirming tight coordination between transcriptional regulation and metabolic flux. KEGG enrichment analysis showed that pathways directly relevant to caffeine catabolism, including Purine metabolism (map00230) and Pentose and glucuronate interconversions (map00040), were markedly upregulated, providing metabolic precursors and cofactors for degradation.

In the Metabolic Homeostasis Stage (22 days), the regulatory focus shifted to energy conservation and metabolite recycling: 69% of TCA cycle DEGs and 72.5% of oxidative phosphorylation DEGs were downregulated, while glycogen synthesis genes were activated. The terminal metabolite xanthine entered the purine metabolic cycle, serving as an alternative energy source and completing the “degradation-recycling” loop. This stage transition reflects a resource reallocation strategy, where nonessential pathways (e.g., amino sugar metabolism, glycerophospholipid metabolism) were persistently suppressed (93 consistently downregulated genes) to prioritize energy and materials for core degradation processes.

### 4.3. Comparative Advantages of D. tabescens in Caffeine Degradation

Compared with other caffeine-degrading microorganisms, *D. tabescens* exhibits distinct functional advantages. Unlike *Pleurotus ostreatus*, which only degrades caffeine under solid-state fermentation [13], *D. tabescens* achieves efficient degradation in liquid culture (85% caffeine removal rate by day 24 at 900 mg/L caffeine), making it more suitable for large-scale applications such as industrial decaffeination and wastewater treatment. Compared with bacterial degraders (e.g., *Pseudomonas putida*) [8], *D. tabescens*’s branched pathway and stage-specific energy regulation reduce energy waste: while bacteria rely on continuous high-energy consumption via fixed gene clusters, *D. tabescens* employs demand-based energy allocation, activating energy pathways only during peak degradation and conserving resources in late stages. Among fungal degraders, *Aspergillus sydowii* [14] and *Penicillium* spp. [21] utilize linear N-demethylation pathways dependent on NADPH-dependent monooxygenases, lacking the branched network and CYP450 partitioning observed in *D. tabescens*. The broad substrate adaptability and reaction specificity of *D. tabescens*’s CYP450 system enable it to accommodate structural variations in caffeine intermediates while minimizing unnecessary energy expenditure, reflecting an evolutionary adaptation to fluctuating caffeine concentrations in natural habitats.

## 5. Conclusions and Future Outlook

This study confirms *Desarmillaria tabescens* as a potent caffeine-degrading fungus, which efficiently metabolizes caffeine under optimized conditions (malt extract medium, 900 mg/L caffeine, 28 °C, pH 8) via a unique branched “N-demethylation–C8 oxidation” network. Transcriptomic analysis identifies nine stage-specific upregulated CYP450 genes as core catalytic components and uncovers a three-phase “Stress-Degradation-Homeostasis” regulatory model that coordinates detoxification, energy metabolism, and secondary metabolism. These findings expand fungal caffeine-degrading resources and reveal a novel metabolic strategy in basidiomycetes.

Future research will first prioritize mechanistic validation of the core CYP450 pathway: employing CRISPR-mediated knockout to confirm the demethylation sequence, conducting in vitro expression of CYP450 proteins to verify substrate specificity, and applying subcellular fractionation to validate peroxisome-associated metabolism. Additionally, while intracellular CYP450 pathways are central to caffeine degradation, transcriptomic data provide preliminary evidence for the potential complementary role of extracellular oxidative enzymes [34]—modest upregulation of laccase-related (log_2_FC = 1.5–2.0, e.g., gene_12345) and peroxidase-related (log_2_FC = 1.3–1.8, e.g., gene_16789) genes was detected under caffeine stress, and these genes share sequence homology with *Pleurotus ostreatus*’s caffeine-degrading laccases [13], suggesting they may target late-stage intermediates (e.g., xanthine, 1,3-dimethyluric acid) to prevent extracellular accumulation. Thus, future work should quantify the catalytic activity of these extracellular enzymes [35,36], test their ability to degrade caffeine intermediates, and clarify whether they form a synergistic network with intracellular CYP450s [37].

We note that our transcriptomic data reflects gene expression trends rather than actual protein abundance, leaving room for post-transcriptional regulation; unquantified enzyme activity (e.g., CYP450 catalytic efficiency) may also cause discrepancies between gene upregulation and metabolic flux, while accumulated intermediates could trigger feedback regulation of degradation-related genes. These regulatory layers will be explored in future work via proteomics and enzyme activity assays. Beyond pathway validation, integrated multiomics will further refine details of the caffeine degradation network. Applied research directions include optimizing *D. tabescens*’s fermentation conditions for bioremediation of caffeine-contaminated waste and development of low-caffeine functional foods, as well as exploring strain engineering or microbial consortia construction to enhance degradation efficiency, thereby unlocking the strain’s full potential for environmental sustainability and industrial applications.

## Figures and Tables

**Figure 1 microorganisms-13-02720-f001:**
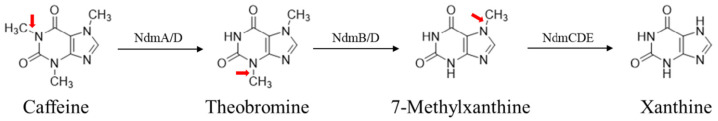
Caffeine N-demethylation degradation pathway. Note: The red arrows label the positions in the molecule that are biochemically attacked upon employing known degradation pathways.

**Figure 2 microorganisms-13-02720-f002:**
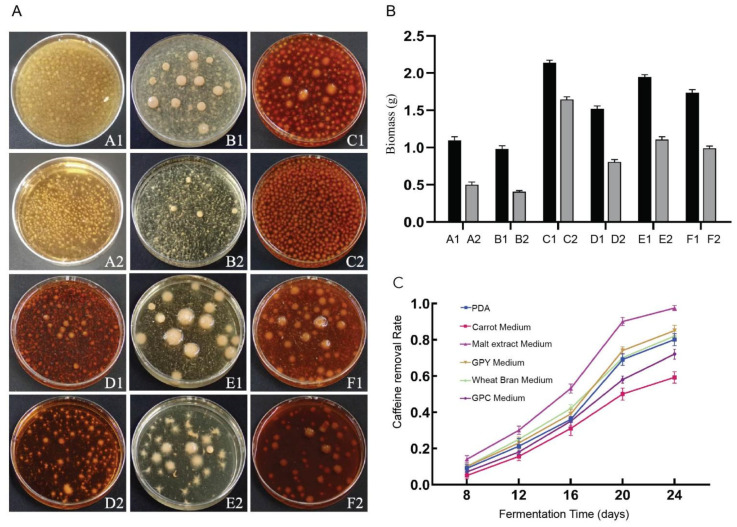
(**A**) Growth of *D. tabescens* on six liquid media. (**B**) Statistics of mycelial biomass (dry weight) of *D. tabescens* in six liquid culture media. (**C**) Statistics on Caffeine Degradation by *D. tabescens* in Six Liquid Culture Media. Note: A1: PDA medium; A2: PDA caffeine enriched medium; B1: carrot medium; B2: carrot caffeine enriched medium; C1: malt extract medium; C2: malt extract caffeine enriched medium; D1: GPY medium; D2: GPY caffeine enriched medium; E1: wheat bran medium; E2: wheat bran caffeine enriched medium; F1: GPC medium; F2: GPC caffeine enriched medium.

**Figure 3 microorganisms-13-02720-f003:**
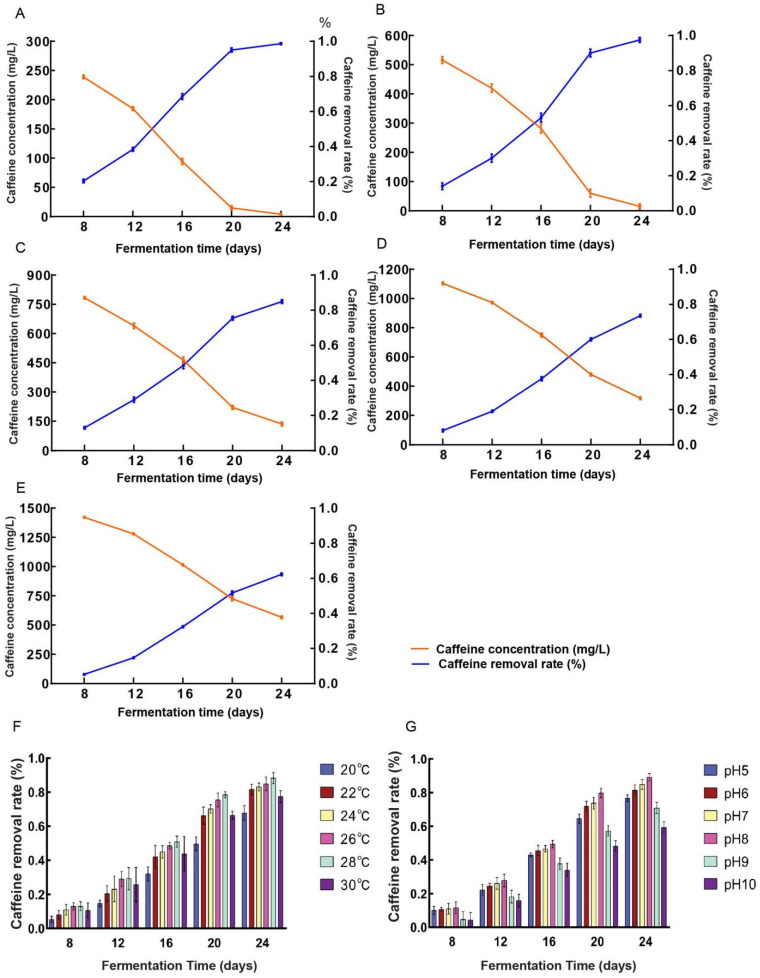
(**A**–**E**) Caffeine degradation by *D. tabescens* in different caffeine concentration (mg/L). Caffeine concentration (**A**) 300 mg/L; (**B**) 600 mg/L; (**C**) 900 mg/L; (**D**) 1200 mg/L; (**E**) 1500 mg/L. (**F**) Caffeine removal rate by *D. tabescens* at different temperature conditions. (**G**) Caffeine removal rate by *D. tabescens* under different pH conditions.

**Figure 4 microorganisms-13-02720-f004:**
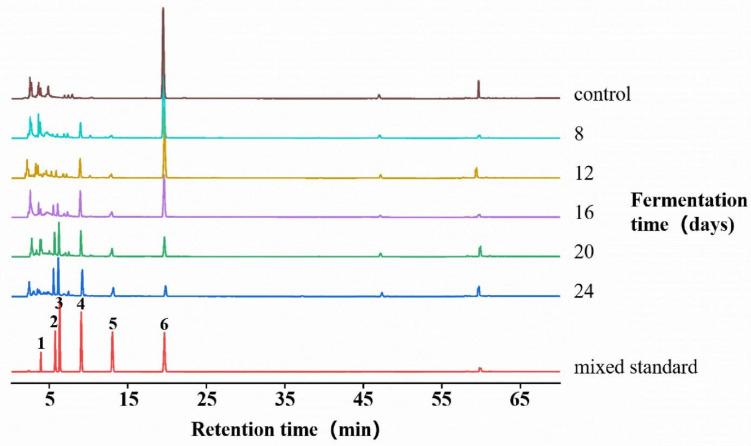
Calibration of caffeine and its metabolites in *D. tabescens* at different fermentation times. Note: 1. xanthine; 2: 7-methylxanthine; 3: 3-methylxanthine; 4: theobromine; 5: theophylline; 6: caffeine.

**Figure 5 microorganisms-13-02720-f005:**
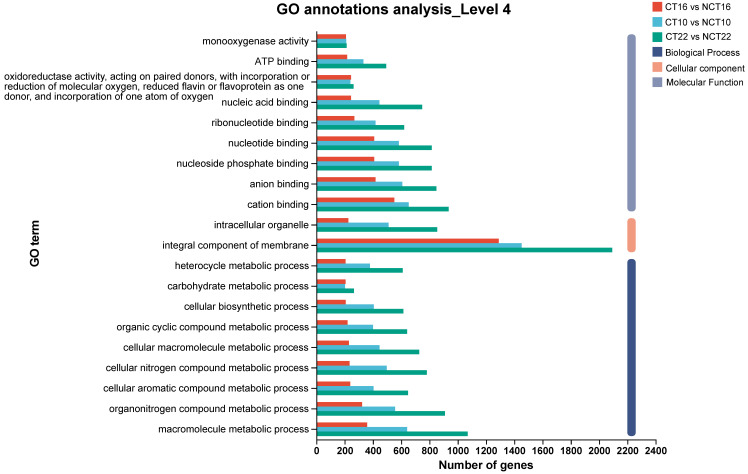
GO classification of DEGs. The *x*-axis indicates the number of DEGs, and the *y*-axis indicates the specific GO term.

**Figure 6 microorganisms-13-02720-f006:**
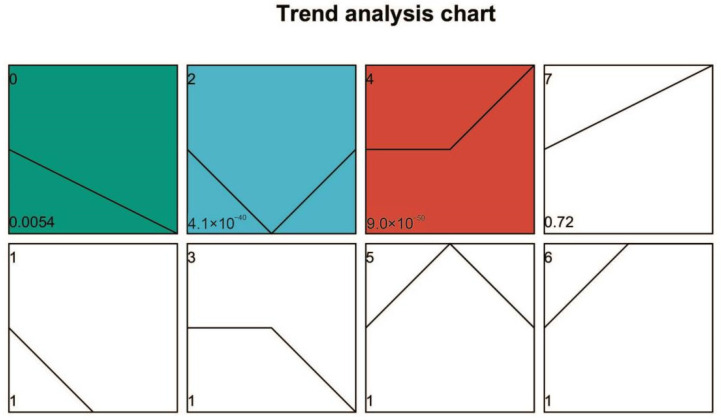
Trend analysis chart of DEGs by STEM. Note: Each profile is represented by a rectangle, with the profile number (starting from 0) in the top-left corner. The line inside the rectangle represents the expression trend over time, and the *p*-value indicating significance is displayed in the bottom-left corner. Colored Trend Profiles: These indicate that the temporal pattern of the profile follows a significant trend. Profiles with the same color belong to the same cluster (profiles with similar trends are grouped together). Uncolored Trend Profiles: These indicate that the temporal pattern of the profile is statistically non-significant.

**Figure 7 microorganisms-13-02720-f007:**
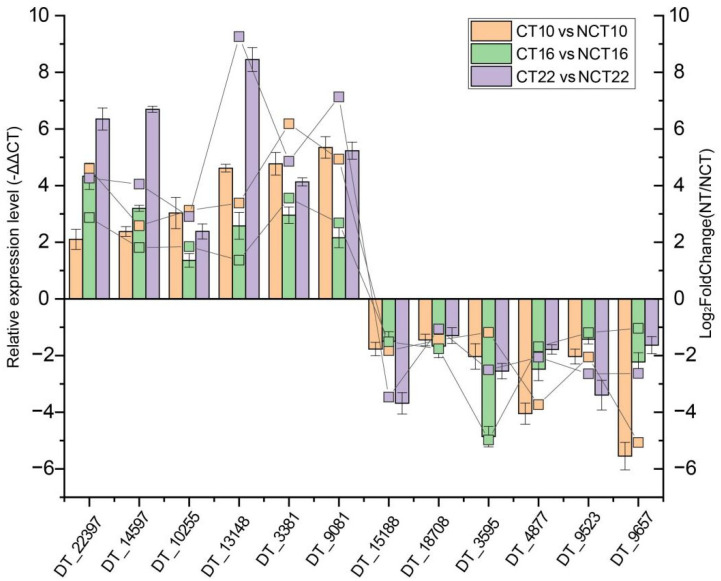
Expression profile validation of 12 selected DEGs by qRT-PCR analysis. The left *y*-axis indicates relative expression levels quantified by qRT-PCR, with error bars representing the mean deviations of three replicates. The right *y*-axis depicts transcript abundance derived from DEG analysis.

**Table 1 microorganisms-13-02720-t001:** Sequence list of primers for qPCR analysis of differentially expressed genes in *D. tabescens*.

Primer	Sequence (5′-3′)	Product Size (bp)
β_tub for	CTGGTTTCGCACCTTTGA	241 bp
β_tub rev	TTGTTGGGAATCCACTCG	
DT_22397-F	TTGCGTTTCTTATTGCGGGATCAG	75 bp
DT_22397-R	GCGTGCGTGCCAGGTAGTAG	
DT_14597-F	ATAGTGCGGCTGTGGCGAAG	148 bp
DT_14597-R	GCTTCATCAGGATTCTCCAACACG	
DT_10255-F	GAATTGAAGGATGGAGTGGCAGAC	138 bp
DT_10255-R	TAGAGGCAGGAAGGCGAAGTTG	
DT_13148-F	ACGGTTTCACACTCTCGCTTAATG	132 bp
DT_13148-R	GCACATCAGATTCCACAGAGACATC	
DT_3381-F	CCGCATGTACGCAGTTATGTCAG	119 bp
DT_3381-R	AGCCTTCCATCACGACCTGTC	
DT_9081-F	GGCGTCCTCTTCTCCGATAACC	125 bp
DT_9081-R	GTGACATTGAGCGAAGCGAAGG	
DT_15188-F	ATGGTCTGGCGGGTCAAAGG	101 bp
DT_15188_R	GCAAGTGTGGCGGATCGTAAC	
DT_18708-F	TTTCCCTTGCCTGCCTCCTC	114 bp
DT_18708-R	GTCGTCGGTTCATCGCTTGG	
DT_3595-F	GCTTGTGCGGCTTACCTTCTAC	86 bp
DT_3595-R	ATCCCTGTGACATGACCCTTGG	
DT_4877-F	GCTCTCAGTACCCGCCCTTTG	87 bp
DT_4877-R	CGATCCGCCAGTCCGTTCTC	
DT_9523-F	GCAAGACCAACAGGATGGATACG	116 bp
DT_9523-R	GCGGCGAAGCGAAGACATAG	
DT_9657-F	TGGCAACAACAGCAAGCAGAAG	124 bp
DT_9657-R	GTGACCATCCGCTATACCTCCTAC	

## Data Availability

The original contributions presented in this study are included in the article/Appendix A. Further inquiries can be directed to the corresponding authors.

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
