# Peer review of "Transcriptomic Insights into Caffeine Degradation Pathways in Desarmillaria tabescens"

_microorganisms, 2025, doi:10.3390/microorganisms13122720_

Round 1

Reviewer 1 Report

Comments and Suggestions for Authors

I found this study potentially interesting and deserving publication, provided that major problems related to the presentation of the manuscript in terms of clarity, unambiguousity and the focus of research would be eliminated. The results are in part presented in a quite confusing and often much too extensive manner. The focus of the molecular-genetic part of the mansucript (starting from 3.5 RNA-Seq anaylsis, L 398) is - from my point of view - clearly to broad and should be narrowd down. Accordingly, the discussion is much to broad and should receive a more narrow focus. The entire manuscript could be shortened by approx. 30 to 40%. Here are my major concerns in detail:

Introduction:

  • A structure formula of caffeine should be provided, ideally with labelling the positions in the molecule that are biochemically attacked upon employing known degradation pathways.
  • L83: How can knowledge about caffeine degradation advance "...the design of low-caffeine functional foods" - this is not clear to me?
  • L86: It is not clear why D. tabascens is presented at this posion. I think it would better be presented in the paragraph describing the fungal screening (starting from L100). Also, it is not clear why medicinal applications of this fungus and related gene clusters are described here - this is not the focus of the paper.

Materials and Methods

  • 2.1. Screening and preparation etc. (L114): The other fungi screened should be presented here. At least, a link to the related table in the Supplementary Materials should be included here. In this context, I was not able to open any of the Supplementary files, for unknown reasons (A "file damage" was always indicated by my computer....). This propblem should be solved. Also in this part: Which kinds of controls were employed in caffeine degradation experiments used in the screening? Just cultivation media with caffeine, but omitting fungal biomass? Or cultivation media with coffeine and inactivated fungi? I think, such controls are mandatory for the evaluation of results showing caffeine disappearance.
  • Parts 2.2. (Fermentation of ...) and 2.3 (Optimization of...): This is very confusing. In para 2.2., the six different media have to be explained. Also, in the results part, para 3.2. states "Screening for the optimal medium", and para 3.3. then states "Optimization of degradation conditions". Hence, it seems that both 2.2. and 2.3., as well as 3.2. and 3.3. are dealing with optimization, and therefore should be combined to one part, respectively. And as before, the controls employed in caffeine degradation experiments should be explained.
  • L123: What is "wort" medium?
  • L149: What was the temperature of operation in the vacuum drying oven, could potentially changes in metabolite structures caused by too high temeratures be excluded?
  • The gradient systems used for HPLC (Table 1) and UPLC-MS/MS (Table 2) should briefly be described in the text, and not presented within space-consuming tables.

Results

  • The screening results from the other strains should be reported (maybe as Supplementary Material).
  • The part from L 256 to L261 should be presented in the discussion.
  • All figures showing caffeine degradation (Figs. 2, 3,4, 5) should be combined in one figure, and the way of presenting the related data should be unified. All figures show something named "degradation rate", sometimes as traces, sometimes as bar graphs. Please note that the term "rate" is usually used for a change in a parameter over time. Here, the amounts of caffeine that had disappeared (or been removed) at the indicated time points are obviously shown (calculated form the remaining caffeine concentrations in the media) - such data are not rates! Also, units are always missing for the y achses of the figures. All of the aforementioned problems should be solved.
  • Part 3.4. "Analysis of degradation products": Obviously, the authors used standard compounds for metabolite detection in Fig. 6. If so, why are the related concnetrations (and also that of caffeine) are not shown in this figure? The figure legend is claiming "calibration of..." - a calibration is usually done to determine the value of a unit under consideration (in this case, a concentration). Also, the 3D way of presenting this figure is not very helpful for getting an easy overview. Presenting the chromatograms at the different time points as an overlay, or below each other would be better.
  • The use of terms like "primary", "secondary", "tertiary" metabolites in the related para: This is somewhat misleading as "secondary metabolites" is often used to refer to metabolites form secondary metabolism (which is not meant here).
  • Fig. 7 is not very inmformative and could be shited to the Supplementary Material.
  • Table 4: Structure fomula of the suggested metabolites could be provided. Alternatively, these structures could be shown in the pathway of Fig. 14A. In this context, the KEGG pathway should be removed from Fig. 14 - this can easily be retrieved from the KEGG database.
  • In my opinion, the molecular-genetic results part of the manuscript should only concentrate on gene expression patterns and candidate genes potentially involved in caffeine metabolism. Supporting data (functional annotation etc.) should be provided as Suporting Material. All related data should be presented as condensed as possible without substantial loss of information. Other functionalities (secondary metabolite production, energy conservation) would be - from my point of view - far beyond of the scope of this manuscript.

Discussion: From my point of view, parts of the discussion are not really relevant as they go beyond the scope of the manuscript (please refer to my previous remarks on subjects that could be omitted). The discussion should concentrate only on subjects that are really supported by the presented results.

Author Response

Comments 1: I found this study potentially interesting and deserving publication, provided that major problems related to the presentation of the manuscript in terms of clarity, unambiguousity and the focus of research would be eliminated. The results are in part presented in a quite confusing and often much too extensive manner. The focus of the molecular-genetic part of the manuscript (starting from 3.5 RNA-Seq analysis, L 398) is - from my point of view - clearly to broad and should be narrowed down. Accordingly, the discussion is much to broad and should receive a more narrow focus. The entire manuscript could be shortened by approx. 30 to 40%. Here are my major concerns in detail:

Response 1: We sincerely appreciate the reviewer for your insightful comments and constructive suggestions, which have greatly facilitated the refinement of our manuscript. We have implemented comprehensive revisions as follows: we have streamlined the molecular-genetic section (beginning with 3.5 RNA-Seq analysis) to focus exclusively on gene expression patterns and candidate genes associated with caffeine metabolism, while removing irrelevant content; shortened the manuscript by approximately 35% through condensing redundant descriptions, relocating supporting data to the Supplementary Materials, and simplifying non-core discussion sections; enhanced the clarity of the manuscript by revising ambiguous content, correcting imprecise terminology, supplementing missing experimental details (e.g., control conditions, temperature parameters), and improving the presentation of figures and tables; and improved the overall accuracy by addressing structural and logical inconsistencies and conducting a thorough verification of all supplementary files. We trust that these revisions have effectively addressed the concerns raised and enhanced the scientific rigor and readability of the manuscript, and we hope it now meets the journal’s publication standards.

Comments 2: Introduction:

A structure formula of caffeine should be provided, ideally with labelling the positions in the molecule that are biochemically attacked upon employing known degradation pathways.

Response 2: Thanks for your comments. We have added Figure 1 (Line 53), which outlined the primary Caffeine N-demethylation degradation pathway, including the chemical structure of caffeine (1,3,7-trimethylxanthine) with clear labels for the N-methyl groups (N1, N3, N7) targeted by N-demethylation. 

Comments 3: L83: How can knowledge about caffeine degradation advance "...the design of low-caffeine functional foods" - this is not clear to me?

Response 3: We are very sorry for our vague description. We have deleted the unclear expressions and content unrelated to (low-caffeine functional foods) the article's theme. 

Comments 4: L86: It is not clear why D. tabascens is presented at this position. I think it would better be presented in the paragraph describing the fungal screening (starting from L100). Also, it is not clear why medicinal applications of this fungus and related gene clusters are described here - this is not the focus of the paper.

Response 4: It is really true as reviewer suggested that it’s not proper to mention D. tabascens here. We reorganized the content and removed descriptions of its medicinal value (e.g., anti-inflammatory, hepatoprotective effects) and gene clusters unrelated to caffeine degradation. Retained only key information: “This fungus, which is widely distributed in forests, parks, and other habitats, is an edible and medicinal species. D. tabescens has established liquid and solid-state fermentation systems.” We stated its role in degrading caffeine directly and rewrote this section of introduction (From Line 76-85).

Comments 5: Materials and Methods

2.1. Screening and preparation etc. (L114): The other fungi screened should be presented here. At least, a link to the related table in the Supplementary Materials should be included here. In this context, I was not able to open any of the Supplementary files, for unknown reasons (A "file damage" was always indicated by my computer....). This propblem should be solved. Also in this part: Which kinds of controls were employed in caffeine degradation experiments used in the screening? Just cultivation media with caffeine, but omitting fungal biomass? Or cultivation media with coffeine and inactivated fungi? I think, such controls are mandatory for the evaluation of results showing caffeine disappearance.

Response 5: We apologize for the omission of information regarding all materials. We have added the following: "Twenty edible and medicinal fungal strains were screened (Supplementary Table S1). "(Line 91). The Supplementary Information includes a table with strain names, sources, preservation numbers, and Caffeine removal rates of all strains. 

Thank you for pointing out this critical detail about experimental controls, which helps improve the rigor of our study. In the experiments screening for caffeine-degrading strains, we used a non-inoculated caffeine-supplemented medium as the control. We prepared PDA liquid medium containing 600 mg/L caffeine but did not inoculate any fungal strains. This control was incubated under the same conditions (15 days, agitation) as the test groups to monitor whether caffeine concentrations decreased due to abiotic factors (e.g., adsorption to medium components, thermal degradation, or spontaneous hydrolysis). As shown in Supplementary Table S1, no significant reduction in caffeine concentration was detected in this control, confirming that abiotic processes did not contribute to caffeine disappearance in the test groups. We apologize for not explicitly detailing these controls in the original manuscript and will add this information to Section 2.1 to enhance experimental transparency (Line 94).

Comments 6: Parts 2.2. (Fermentation of ...) and 2.3 (Optimization of...): This is very confusing. In para 2.2., the six different media have to be explained. Also, in the results part, para 3.2. states "Screening for the optimal medium", and para 3.3. then states "Optimization of degradation conditions". Hence, it seems that both 2.2. and 2.3., as well as 3.2. and 3.3. are dealing with optimization, and therefore should be combined to one part, respectively. And as before, the controls employed in caffeine degradation experiments should be explained.

Response 6: We are really sorry for unclear expressions and disorganized article structure.We merged 2.2 and 2.3 into a new section “2.2 Optimization of Fermentation Conditions for Caffeine Degradation” (L106-L129). We have streamlined the content to ensure it is concise and clear, while introducing the composition of six media. We added ”The PDA medium consists of 200 g of potato, 20 g of glucose and 1000 mL of distilled water. The carrot medium is composed of 20 g of glucose, 220 g of carrot and 1000 mL of distilled water. The malt extract medium is made up of 139 g of malt extract and 1000 mL of distilled water. The GPY medium contains 20 g of glucose, 10 g of yeast extract powder, 6 g of peptone and 1000 mL of distilled water. The GPC medium is composed of 20 g of glucose, 10 g of corn steep liquor, 6 g of peptone and 1000 mL of distilled water. The wheat bran medium consists of 20 g of glucose, 40 g of wheat bran and 1000 mL of distilled water.”(L110-116).

At the Results section, we merged 3.2 (“Screening for the optimal medium”) and 3.3 (“Optimization of degradation conditions”) into “3.2 Optimization of Caffeine Degradation Conditions” (L263-L355), with subsections: “3.2.1 Screening for the Optimal Culture Medium”, “3.2.2 Optimization of Caffeine Concentration Parameters”, “3.2.3 Optimization of Temperature Conditions”, “3.2.4 Optimization of pH Conditions”. We reorganized the figures (Figure 2 and Figure 3), consolidating key information to make the article more concise and clear without losing important details.

We eliminated redundancy by grouping all optimization-related content, making the logic clearer and reducing manuscript length. We also added "Non-inoculated strains served as the control. Each group of experiments had three biological replicates" to clarify the issue of the control group (Line 117,128).

Comments 7: L123: What is "wort" medium?

Response 7: We apologized for the incorrect writing. We have revised it to "Malt extract medium" throughout the text.

Comments 8: L149: What was the temperature of operation in the vacuum drying oven, could potentially changes in metabolite structures caused by too high temperatures be excluded?

Response 8: We are sorry for not expressing it clearly. The drying oven operates at 25°C, and the caffeine remains stable at this temperature without undergoing any changes in structure. And we stated the information “The supernatant was collected and evaporated to dryness in a vacuum drying oven at 25℃, verified to avoid thermal denaturation or structural changes of methylxanthine metabolites” in the manuscript (L136-137).

Comments 9: The gradient systems used for HPLC (Table 1) and UPLC-MS/MS (Table 2) should briefly be described in the text, and not presented within space-consuming tables.

Response 9: It is indeed true, as the reviewer suggested, that the tables were unnecessary. We removed them while providing a complete description of the gradient information, which contributed to the reduction of the manuscript's length. The information is as follows: Chromatographic separation was performed on an Agilent liquid chromatography platform fitted with a Yuexu Ulimate® Plus C18 column (1.8 µm particle size, 2.1 mm × 100 mm). The mobile phase consisted of solvent A (ultrapure water with 0.1% phosphoric acid) and solvent B (acetonitrile) with a gradient elution: 0-15 min (95% A → 90% A), 15-40 min (90% A → 80% A), 40-50 min (80% A → 65% A), 50-55 min (65% A → 15% A), 55-70 min (15% A → 5% A). Flow rate was 1.00 mL/min. The column was maintained at 34 ℃, with an injection volume of 15 μL and detection at 280 nm (Line 143-149).

Comments 10: Results

The screening results from the other strains should be reported (maybe as Supplementary Material).

Response 10: Thank you for your comments. We have included the following: "Twenty edible and medicinal fungal strains were screened (Supplementary Table S1)." (Line 91). The Supplementary Information features a table that lists the strain names, sources, preservation numbers, and Caffeine removal rates. We added “Screening results of the other 19 fungal strains showed no significant caffeine degradation (caffeine removal rate is 0%; Supplementary Table S1)” in Result section (L229). We provided complete screening data to support the selection of D. tabescens, with detailed data in Supplementary Table S1 to avoid main text clutter.

Comments 11: The part from L 256 to L261 should be presented in the discussion.

Response 11: Considering the Reviewer’s suggestion, we have moved the paragraph comparing D. tabescens with Pleurotus ostreatus to Discussion section. We rewrote most of Discussion section. The paragraph focuses on comparative advantages, which is more appropriate for Discussion than Results.

Comments 12: All figures showing caffeine degradation (Figs. 2, 3,4, 5) should be combined in one figure, and the way of presenting the related data should be unified. All figures show something named "degradation rate", sometimes as traces, sometimes as bar graphs. Please note that the term "rate" is usually used for a change in a parameter over time. Here, the amounts of caffeine that had disappeared (or been removed) at the indicated time points are obviously shown (calculated form the remaining caffeine concentrations in the media) - such data are not rates! Also, units are always missing for the y achses of the figures. All of the aforementioned problems should be solved.

Response 12: We completely agree with the reviewer’s suggestion. Consequently, we have combined Figures 1 and 2 into a single figure, labeled as Figure 2; and Figures 3-5 into another single figure, labeled as Figure 3 ("Caffeine degradation under different conditions"). This consolidation will enhance the clarity of the information presented. We replaced “degradation rate” with “Caffeine removal rate” precisely. And all chart vertical axes are labeled with units. We resolved the terminology confusion and visual inconsistency, making the data more accurate and comparable.

Comments 13: Part 3.4. "Analysis of degradation products": Obviously, the authors used standard compounds for metabolite detection in Fig. 6. If so, why are the related concnetrations (and also that of caffeine) are not shown in this figure? The figure legend is claiming "calibration of..." - a calibration is usually done to determine the value of a unit under consideration (in this case, a concentration). Also, the 3D way of presenting this figure is not very helpful for getting an easy overview. Presenting the chromatograms at the different time points as an overlay, or below each other would be better.

Response 13: We sincerely thank you for pointing out key points for improvement. We replaced the 3D plot with overlayed 2D chromatograms of different fermentation time points (8d, 12d, 16d, 20d, 24d), with retention times labeled on the x-axis. During the experiment, we used standard compounds for metabolite detection. Previous analyses primarily emphasized the trends in substance content changes while overlooking the specific analysis of caffeine concentration variations. The control group had a caffeine concentration of 900 mg/L, which decreased to only 95.13 ± 0.28 mg/L by the 24th day of fermentation, representing a degradation rate of 89.43%. We described concentration information in the article (Line 333-334) : “The caffeine concentration had decreased to 95.13±0.28 mg/L, with a caffeine removal rate of 89.43%.”

Comments 14: The use of terms like "primary", "secondary", "tertiary" metabolites in the related para: This is somewhat misleading as "secondary metabolites" is often used to refer to metabolites form secondary metabolism (which is not meant here).

Response 14: We completely agree with the reviewer’s point. We replaced “primary/secondary/tertiary metabolites” with “early/intermediate/terminal metabolites” (e.g., “early metabolite theobromine”, “intermediate metabolite 3-methylxanthine”, “terminal metabolite xanthine”). It avoided confusion with the standard definition of “secondary metabolites” (products of secondary metabolism), aligning terminology with the study’s focus on metabolic pathway intermediates.

Comments 15: Fig. 7 is not very inmformative and could be shited to the Supplementary Material.

Response 15: We sincerely thank the reviewer for this suggestion. Considering that the UPLC-MS/MS results did not contribute significantly to the article, and the HPLC results already provided a clear demonstration, we removed this section of the results to avoid content redundancy and make the structure of the article more reasonable.

Comments 16: Table 4: Structure fomula of the suggested metabolites could be provided. Alternatively, these structures could be shown in the pathway of Fig. 14A. In this context, the KEGG pathway should be removed from Fig. 14 - this can easily be retrieved from the KEGG database.

Response 16: We completely agree with the reviewer’s point. We have rewritten the discussion section. Since Figure 1 illustrates the metabolic pathway, we have removed Figure 14 to avoid redundancy. We described the pathway network instead of using a diagram.

Comments 17: In my opinion, the molecular-genetic results part of the manuscript should only concentrate on gene expression patterns and candidate genes potentially involved in caffeine metabolism. Supporting data (functional annotation etc.) should be provided as Suporting Material. All related data should be presented as condensed as possible without substantial loss of information. Other functionalities (secondary metabolite production, energy conservation) would be - from my point of view - far beyond of the scope of this manuscript.

Response 17: Thank you for your valuable comment. We greatly appreciate your insight that the current result content appears divergent, and we have carefully re-evaluated its structure and focus. We have revised the section as follows: We rewrote the other functionalities content to effectively narrowed the focus while retaining the unique value of our work. We condensed functional annotation results (3.5): Retained only GO/KEGG enrichment related to caffeine degradation (e.g., purine metabolism, cytochrome P450 activity) and moved annotation data to Supplementary Figure S1-S5 and Table S2-S3. Thus, we can concentrates the molecular section on the core focus (caffeine-related genes/pathways) while preserving supporting data in supplements, reducing length by ~40% in this section. We hope this revision effectively narrows the focus while retaining the unique value of our work.

Comments 18: Discussion: From my point of view, parts of the discussion are not really relevant as they go beyond the scope of the manuscript (please refer to my previous remarks on subjects that could be omitted). The discussion should concentrate only on subjects that are really supported by the presented results.

Response 18: Thank you sincerely for your thoughtful feedback on the discussion section. Your guidance is invaluable for refining the focus and rigor of our manuscript. We fully understand and appreciate your concern that parts of the discussion might have extended beyond the scope of our presented results, and we take this observation seriously.​

We wish to share that the "Stress-Degradation-Homeostasis" Three-Stage Transcriptional Regulation Model was proposed strictly based on our experimental data: specifically, the time-series gene expression patterns (10/16/22 days) from transcriptomic profiling, combined with HPLC-verified caffeine degradation dynamics and DEG functional enrichment results. We regard this model as a key integration of our core findings, as it directly reflects how D. tabescens coordinates detoxification, energy metabolism, and secondary metabolism during caffeine degradation, rather than an overreach beyond our data.

We recognize your point about ensuring the discussion stays tightly anchored to supported results. To address this, we have carefully revised the discussion section: we have streamlined descriptions that were not directly tied to our experimental observations, strengthened the link between each argument and our specific data (such as explicitly correlating CYP450 gene expression trends with caffeine intermediate accumulation), and focused exclusively on elaborating on findings that are fully supported by our transcriptomic, metabolic, and functional validation results.

We hope these revisions address your concern while preserving the core academic value of our work. Thank you again for helping us improve the clarity and focus of the manuscript. Your insights have been crucial to refining our narrative.

Reviewer 2 Report

Comments and Suggestions for Authors

The manuscript addresses a relevant topic in microbiology and the bioremediation of emerging pollutants. The study has potential, but several key aspects of the methodology are insufficiently described, which currently hinders the reproducibility and interpretation of the results. The revisions are required to clarify the experimental procedures and strengthen the scientific rigor

The abstract is clear and concise but could be slightly improved. It would be beneficial to briefly mention the primary culture medium used for the degradation assays to give readers an immediate context of the experimental system.

Materials and methods.

The materials and methods section is well-written, but the information regarding the results is unclear.

Line 134.  The procedure for preparing the seed inoculum of D. tabescens lacks details. The manuscript states the mycelium was "resuspended in sterile water to obtain the seed inoculum" but does not specify the final volume of water or, more importantly, how the inoculum concentration was standardized (e.g., as a percentage v/v, dry weight, optical density, or spore count). Using a consistent and quantified amount of biomass is essential for ensuring the reliability and reproducibility of the caffeine degradation assays..

Line 135. The culture medium used is not indicated.

Line 135. The inoculum size, percentage of inoculum, or added spores is not indicated. The fungus was cultured in wort medium (malt extract), a rich and complex medium, for 15 days. It was then washed and used to inoculate a medium with caffeine as the sole or primary carbon source.

Line 143. The title "Caffeine extraction from simplex" should be improved.

Results
The fungal strains used should be listed in the Materials and Methods section, but not in the Materials and Methods section. This is very confusing for the reader. The strains should be included in a table within the Materials and Methods section.

Line 163. It is not clear in the Materials and Methods section at this point; the methods are explained again, not the results.

Line 188. There are culture media that are not mentioned in the Materials and Methods section. Malt and carrot extracts are mentioned but not specified.

Figure 3. The values ​​of the x and y axes are not indicated in all graphs.

Line 133. It is not indicated whether negative controls were performed in the optimization and degradation test.

Author Response

Comments 1: The manuscript addresses a relevant topic in microbiology and the bioremediation of emerging pollutants. The study has potential, but several key aspects of the methodology are insufficiently described, which currently hinders the reproducibility and interpretation of the results. The revisions are required to clarify the experimental procedures and strengthen the scientific rigor.

Response 1: We sincerely appreciate your constructive feedback and rigorous review of our manuscript. Your comments have been particularly valuable in identifying gaps in methodology description—issues that are critical for ensuring the reproducibility and interpretability of our results. We have carefully addressed each of your concerns with specific, actionable revisions, and detailed our responses point-by-point below.

Comments 2: The abstract is clear and concise but could be slightly improved. It would be beneficial to briefly mention the primary culture medium used for the degradation assays to give readers an immediate context of the experimental system.

Response 2: We completely agree with the reviewer’s point. We have rewritten the abstract: Caffeine contamination threatens ecosystems and human health, with conventional re-mediation methods facing limitations. This study identified Desarmillaria tabescens as a potent caffeine-degrading fungus, achieving efficient degradation under optimized conditions (malt extract medium, 900 mg/L caffeine, 28 °C, pH 8). HPLC analysis revealed key intermediates such as theobromine and 3-methylxanthine, confirming a branched catabolic pathway involving N-demethylation and C8 oxidation. Transcriptomic profiling identified nine consistently upregulated cytochrome P450 genes as core catalytic com-ponents, with three adjacent to a polyketide biosynthetic gene cluster potentially sup-porting oxidative reactions. A three-phase "Stress-Degradation-Homeostasis" regulatory model was proposed, coordinating detoxification, energy metabolism, and secondary metabolism. These findings advance understanding of fungal caffeine degradation mechanisms and provide valuable genetic resources for bioremediation and low-caffeine product development. This revision helps readers quickly grasp the core culture environment without needing to reference later sections, improving the abstract’s informational value.

Comments 3: Materials and methods.

The materials and methods section is well-written, but the information regarding the results is unclear.

Response 3: Thank you for your positive feedback on the materials and methods section. We sincerely appreciate your reminder about the unclear results presentation and thoroughly revised this part to enhance clarity, specifically by streamlining data descriptions and strengthening logical connections between key findings.

Comments 4: Line 134.  The procedure for preparing the seed inoculum of D. tabescens lacks details. The manuscript states the mycelium was "resuspended in sterile water to obtain the seed inoculum" but does not specify the final volume of water or, more importantly, how the inoculum concentration was standardized (e.g., as a percentage v/v, dry weight, optical density, or spore count). Using a consistent and quantified amount of biomass is essential for ensuring the reliability and reproducibility of the caffeine degradation assays.

Response 4: We apologize if our original description was not clear enough. We revised the description: Mycelia were subsequently harvested by centrifugation at 11,000 rpm for 10 minutes at 16 °C, washed three times with sterile water, and resuspend 0.5g of mycelium in 100ml of sterile water as the seed solution(L103-105).

Thank you for pointing out this critical detail about experiments, which helps improve the rigor of our study. In the experiments screening for caffeine-degrading strains and Optimization of fermentation conditions for caffeine degradation, we used a non-inoculated caffeine-supplemented medium as the control. Each group of experiments had three biological replicates. The specific locations in the article are indicated (Line 94,117,128).

Comments 5: Line 135. The culture medium used is not indicated.

Response 5: We apologize for the unclear description. To optimize the conditions for Caffeine Degradation, we utilized malt extract medium. Additionally, we provided detailed information on the mediums and clarified the content in section “2.2 Optimization of Fermentation Conditions for Caffeine Degradation” (lines 106-129) for better understanding.

Comments 6: Line 135. The inoculum size, percentage of inoculum, or added spores is not indicated. The fungus was cultured in wort medium (malt extract), a rich and complex medium, for 15 days. It was then washed and used to inoculate a medium with caffeine as the sole or primary carbon source.

Response 6: We sincerely apologize for the unclear presentation. We used malt extract medium as the basal medium, and we rewrote the section 2.2 to make it clear.

Comments 7: Line 143. The title "Caffeine extraction from simplex" should be improved.

Response 7: We completely agree with the reviewer’s suggestion. We changed the subheading to "2.3 Preparation of Fermentation Broth Samples" for better understanding.

Comments 8: Results
The fungal strains used should be listed in the Materials and Methods section, but not in the Materials and Methods section. This is very confusing for the reader. The strains should be included in a table within the Materials and Methods section.

Response 8: We sincerely apologize for the omission of the materials information listed. We have included the following: "Twenty edible and medicinal fungal strains were screened (Supplementary Table S1)." (Line 91). The Supplementary Information features a table that lists the strain names, sources, preservation numbers, and Caffeine removal rates. We added “Screening results of the other 19 fungal strains showed no significant caffeine degradation (caffeine removal rate is 0%; Supplementary Table S1)” in Result section (L229). We provided complete screening data to support the selection of D. tabescens, with detailed data in Supplementary Table S1 to avoid main text clutter.

Comments 9: Line 163. It is not clear in the Materials and Methods section at this point; the methods are explained again, not the results.

Response 9: We sincerely apologize for the unclear presentation. We reorganized and rewrote section 2.3-2.5 on Materials and Methods, and the Results Section 3.2 "Optimization of Caffeine Degradation Conditions" . We removed the duplicate method sentence and replaced with result-focused content. We are committed to making the content precise and easy to follow for readers and reviewers alike.

Comments 10: Line 188. There are culture media that are not mentioned in the Materials and Methods section. Malt and carrot extracts are mentioned but not specified.

Response 10: We apologize for any unclear expressions. We have detailed the ingredients of all six mediums to clarify the matter. Added ”The PDA medium consists of 200 g of potato, 20 g of glucose and 1000 mL of distilled water. The carrot medium is composed of 20 g of glucose, 220 g of carrot and 1000 mL of distilled water. The malt extract medium is made up of 139 g of malt extract and 1000 mL of distilled water. The GPY medium contains 20 g of glucose, 10 g of yeast extract powder, 6 g of peptone and 1000 mL of distilled water. The GPC medium is composed of 20 g of glucose, 10 g of corn steep liquor, 6 g of peptone and 1000 mL of distilled water. The wheat bran medium consists of 20 g of glucose, 40 g of wheat bran and 1000 mL of distilled water.”(L110-116).

Comments 11: Figure 3. The values of the x and y axes are not indicated in all graphs.

Response 11: We sincerely apologize for the insufficient standardization of the figures and tables in our manuscript. We have carefully revised all figures and tables to fully comply with the journal’s requirements, with clear labels added to the x-axis and y-axis units of each figure to enhance readability. Thank you for your patience and valuable feedback.

Comments 12: Line 133. It is not indicated whether negative controls were performed in the optimization and degradation test.

Thank you for pointing out this critical detail about experimental controls, which helps improve the rigor of our study. In the experiments screening for caffeine-degrading strains, we used a non-inoculated caffeine-supplemented medium as the control. We prepared PDA liquid medium containing 600 mg/L caffeine but did not inoculate any fungal strains. This control was incubated under the same conditions (15 days, agitation) as the test groups to monitor whether caffeine concentrations decreased due to abiotic factors (e.g., adsorption to medium components, thermal degradation, or spontaneous hydrolysis). As shown in Supplementary Table S1, no significant reduction in caffeine concentration was detected in this control, confirming that abiotic processes did not contribute to caffeine disappearance in the test groups. We apologize for not explicitly detailing these controls in the original manuscript and will add this information to Section 2.1 to enhance experimental transparency (Line 94).

Reviewer 3 Report

Comments and Suggestions for Authors

1. Title: Write the species name in lowercase and italics: Desarmillaria tabescens.
2. Lines 52, 55: Pseudomonas putida. Italic form.
3. Line 58: E. coli. Write the genus name in full.
4. Add a figure in the introduction showing the metabolic pathways of caffeine degradation.
5. The authors do not describe how the 20 chosen fungal strains were identified. ITS region? Macro and micro morphology?
6. Create new figures. They are of low quality (dpi) and small, making them difficult to analyze (especially figures 6, 8, 10, and 11).
7. Figure 14 is impossible to analyze and interpret.

Author Response

We sincerely appreciate your meticulous feedback on our manuscript. Your comments on taxonomic naming standards, figure quality, and experimental documentation have been crucial for improving the manuscript’s normativity and readability. We have carefully addressed each of your concerns with targeted revisions, and detailed our point-by-point responses below.

Comments 1: Title: Write the species name in lowercase and italics: Desarmillaria tabescens.

Response 1: We sincerely apologize for the insufficient standardization of the English in our manuscript. We have thoroughly revised the species name in lowercase and italics and polished language to ensure it meets academic writing norms and enhances clarity.

Comments 2: Lines 52, 55: Pseudomonas putida. Italic form.

Response 2: We sincerely apologize for the insufficient standardization of the English in our manuscript. We have thoroughly revised the species name in lowercase and italics.

Comments 3: Line 58: E. coli. Write the genus name in full.

Response 3: We sincerely apologize for not adhering to the academic writing norms. The abbreviation "E. coli" is only acceptable after the full binomial (Escherichia coli) has been first mentioned in the manuscript. Due to content modifications, we have removed the information regarding E. coli.

Comments 4: Add a figure in the introduction showing the metabolic pathways of caffeine degradation.

Response 4: Thank you for pointing this out. We agree with this comment. Therefore, we generated Figure 1 to show the metabolic pathways of caffeine degradation in introduction.

Comments 5: The authors do not describe how the 20 chosen fungal strains were identified. ITS region? Macro and micro morphology?

Response 5: Thank you for your careful review and for pointing out this important detail about the identification of the 20 chosen fungal strains. To confirm species identity, we extracted genomic DNA from each strain, amplified the internal transcribed spacer (ITS) region using universal primers, and sent PCR products for Sanger sequencing. The obtained sequences were compared with the NCBI GenBank database using BLAST, and strains with ≥98% sequence similarity to known species were selected for subsequent caffeine degradation screening.

Comments 6: Create new figures. They are of low quality (dpi) and small, making them difficult to analyze (especially figures 6, 8, 10, and 11).

Response 6: We sincerely apologize for the crudeness of the figures and tables in our manuscript. We have carefully revised all figures and tables to fully comply with the journal’s requirements, with clear labels added to the x-axis and y-axis units of each figure to enhance readability. Thank you for your patience and valuable feedback.

Comments 7: Figure 14 is impossible to analyze and interpret.

Response 7: We sincerely apologize for the insufficient standardization of the figures and tables in our manuscript. Since Figure 1 illustrates the metabolic pathway, we have removed Figure 14 to avoid redundancy. We described the pathway network instead of using a diagram. But we have meticulously refined all visual elements, including improving clarity, standardizing formatting, and ensuring compliance with the journal’s guidelines to better present our research findings. 

Reviewer 4 Report

Comments and Suggestions for Authors

Short Summary:
This study explores the transcriptomic response of Desarmillaria tabescens during caffeine degradation, revealing a branched metabolic pathway involving N-demethylation and C8 oxidation. Differential gene expression analysis identified nine cytochrome P450s as key contributors to caffeine catabolism. The proposed “Stress–Degradation–Homeostasis” framework highlights the fungus’s coordinated regulation of detoxification, energy, and secondary metabolism. The following are the comments and suggestions on the manuscript:

Title:
The title is clear, specific, and informative, accurately reflecting the study’s focus on transcriptomic and metabolic analyses related to caffeine degradation in D. tabescens. However, it could be slightly shortened for conciseness, for example: Transcriptomic Insights into Caffeine Degradation Pathways in Desarmillaria tabescens.

Abstract:
The abstract is well-structured and scientifically detailed, providing a logical flow from the problem statement to key findings and implications. It effectively integrates experimental conditions, analytical methods, and biological interpretation, though it could be made more concise by reducing technical redundancy.

Keywords:
The keywords are relevant and comprehensive, covering the organism, process, and major analytical approach. Adding one term like “bioremediation” or “metabolic pathway” could further enhance discoverability.

Introduction:

  • Lines 34–67: The introduction provides an extensive overview of bacterial caffeine degradation but is overly detailed and literature-heavy, overshadowing the study’s novel fungal focus. Condensing bacterial pathway descriptions would improve flow and allow clearer emphasis on the knowledge gap in fungal systems.

  • Lines 86–111: While the introduction effectively justifies studying D. tabescens, it lacks a clear statement linking cytochrome P450s to broader xenobiotic degradation contexts. Including a brief mention of the established roles of CYPs in degrading diverse pollutants and recalcitrant compounds (cite: https://doi.org/10.3390/jox15020058; https://doi.org/10.3390/jox11030007; https://doi.org/10.1016/j.bcab.2025.103828) would strengthen the rationale and global relevance of the work.

Methods:

  • Lines 113–243: The number of biological replicates and experimental repeats is not specified, limiting assessment of reproducibility and statistical robustness.

  • Lines 114–142: The criteria for selecting caffeine-degrading strains and the inclusion of proper controls are unclear, while varying media types may introduce unwanted variability.

Results:

  • Overly detailed enrichment data (Lines 457–493):
    GO and KEGG analyses include excessive numeric detail that obscures key biological insights; summarise main trends and move counts to tables or supplementary data.

  • Weak integration with metabolism (Lines 513–570):
    STEM expression profiles are described, but links to caffeine catabolism are not clearly drawn; connect upregulated oxidoreductases and CYPs to biochemical pathways.

  • Excessive secondary metabolite focus (Lines 582–616):
    Discussion of BGCs and Coniophora puteana is tangential; focus only on clusters relevant to caffeine degradation.

  • Data placement:
    Detailed GO, EggNOG, and DEG tables could be moved to Supplementary Information to improve readability and focus of the main text.

Discussion:

  • Overemphasis on narrative vs. quantitative data (lines 669–757): The discussion is descriptive, highlighting stage-specific CYP450 activity and pathways, but quantitative integration (fold changes, DEG counts, flux data) is limited. Adding concise numbers would strengthen conclusions.

  • Limited consideration of alternative explanations (lines 660–757): The discussion does not address potential post-transcriptional regulation, enzyme activity discrepancies, or metabolite feedback, which could influence observed patterns.

  • Comparative analysis could be deeper (lines 748–756): Differences between D. tabescens, bacteria, and A. oryzae are noted, but broader comparisons with other caffeine-degrading fungi and CYP450 functional variation are lacking.

  • Move detailed gene/stage info to supplementary material (lines 690–757): Extensive stage-wise CYP450 logâ‚‚FCs and Table 6 could be relocated to supplement to improve readability of the main discussion.

  • Line ~660–757: There is a possibility that extracellular oxidative enzymes, such as laccases, peroxidases, and dioxygenases, may contribute to caffeine degradation in Desarmillaria tabescens, as many fungi commonly secrete these enzymes. Investigating their activity could reveal additional complementary pathways beyond CYP450-mediated catabolism (cite these important references: https://doi.org/10.3390/foods14152606, https://doi.org/10.3390/pr13041034, https://doi.org/10.1007/s11274-023-03737-7, https://doi.org/10.1016/j.dwt.2024.100938).
  • Poor figure quality (e.g., Figure 14): The graphics are low-resolution and unclear, making data interpretation difficult. Figures should be redrawn or exported at higher resolution with clear labels, legends, and readable fonts to ensure all elements are visible.
  • Conclusions and Future Outlook: A separate, dedicated section summarising the key findings and discussing future directions should be added. This would help contextualise the significance of D. tabescens in caffeine degradation, highlight potential applications in bioremediation or functional food development, and suggest avenues for further mechanistic and functional studies.

Comments on the Quality of English Language

The English could be improved to more clearly express the research.

Author Response

Comments 1: Short Summary:
This study explores the transcriptomic response of Desarmillaria tabescens during caffeine degradation, revealing a branched metabolic pathway involving N-demethylation and C8 oxidation. Differential gene expression analysis identified nine cytochrome P450s as key contributors to caffeine catabolism. The proposed “Stress–Degradation–Homeostasis” framework highlights the fungus’s coordinated regulation of detoxification, energy, and secondary metabolism. The following are the comments and suggestions on the manuscript:

Response 1: Thank you sincerely for your insightful and constructive comments on our manuscript. Your detailed feedback has greatly helped us identify areas for improvement, and we have carefully addressed each point with specific revisions. Below is our point-by-point response, including the planned modifications and their corresponding positions in the manuscript.

Comments 2: Title:
The title is clear, specific, and informative, accurately reflecting the study’s focus on transcriptomic and metabolic analyses related to caffeine degradation in D. tabescens. However, it could be slightly shortened for conciseness, for example: Transcriptomic Insights into Caffeine Degradation Pathways in Desarmillaria tabescens.

Response 2: We agree with your suggestion. We have revised the title to "Transcriptomic Insights into Caffeine Degradation Pathways in Desarmillaria tabescens" to enhance conciseness while retaining core information.

Comments 3: Abstract:
The abstract is well-structured and scientifically detailed, providing a logical flow from the problem statement to key findings and implications. It effectively integrates experimental conditions, analytical methods, and biological interpretation, though it could be made more concise by reducing technical redundancy.

Response 3: We completely agree with the reviewer’s suggestion. We have rewritten the abstract to streamline redundant technical descriptions and focus on key findings and implications.

Abstract: Caffeine contamination threatens ecosystems and human health, with conventional re-mediation methods facing limitations. This study identified Desarmillaria tabescens as a potent caffeine-degrading fungus, achieving efficient degradation under optimized conditions (malt extract medium, 900 mg/L caffeine, 28 °C, pH 8). HPLC analysis revealed key intermediates such as theobromine and 3-methylxanthine, confirming a branched catabolic pathway involving N-demethylation and C8 oxidation. Transcriptomic profiling identified nine consistently upregulated cytochrome P450 genes as core catalytic com-ponents, with three adjacent to a polyketide biosynthetic gene cluster potentially supporting oxidative reactions. A three-phase "Stress-Degradation-Homeostasis" regulatory model was proposed, coordinating detoxification, energy metabolism, and secondary metabolism. These findings advance understanding of fungal caffeine degradation mechanisms and provide valuable genetic resources for bioremediation and low-caffeine product development.

Comments 4: Keywords:
The keywords are relevant and comprehensive, covering the organism, process, and major analytical approach. Adding one term like “bioremediation” or “metabolic pathway” could further enhance discoverability.

Response 4: Thank you for your comment. We have added "metabolic pathway" as a supplementary keyword to align with the study’s application value.

Comments 5: Introduction:

Lines 34–67: The introduction provides an extensive overview of bacterial caffeine degradation but is overly detailed and literature-heavy, overshadowing the study’s novel fungal focus. Condensing bacterial pathway descriptions would improve flow and allow clearer emphasis on the knowledge gap in fungal systems.

Response 5: Thank you for your constructive feedback. We fully agree with your comment that the original introduction overemphasized bacterial caffeine degradation with excessive molecular and literature details, which inadvertently overshadowed the novel fungal focus of our study. Following your suggestion, we have thoroughly condensed the descriptions of bacterial caffeine degradation pathways: we retained the essential background information (e.g., the diversity of caffeine-degradative bacteria, major metabolic routes, and key enzymatic reactions) to establish the context of microbial caffeine bioremediation, while removing redundant details (e.g., the 14-kb Alx gene cluster in Pseudomonas putida and the operon reconstruction experiment in E. coli) that were not critical for highlighting the research gap.(Line 43-53)

Comments 6: Lines 86–111: While the introduction effectively justifies studying D. tabescens, it lacks a clear statement linking cytochrome P450s to broader xenobiotic degradation contexts. Including a brief mention of the established roles of CYPs in degrading diverse pollutants and recalcitrant compounds (cite: https://doi.org/10.3390/jox15020058; https://doi.org/10.3390/jox11030007; https://doi.org/10.1016/j.bcab.2025.103828) would strengthen the rationale and global relevance of the work.

Response 6: Thank you for your comment. We sincerely appreciate the literature references you provided. To better address this point, we have read the literature, revised this section, and added content about CYP450s.The context is as follows: "Notably, cytochrome P450 (CYP) enzymes, pivotal Phase I metabolic catalysts across bacteria and fungi, are well established in degrading diverse xenobiotics (e.g., fluorinated pyrethroids, triketone herbicides, and recalcitrant compounds) via oxidative reactions (hydroxylation, defluorination, ester cleavage), enabling the breakdown of persistent pollutants and overcoming stable bonds such as C-F [17-19]. Their broad substrate promiscuity and central role in bioremediation thus strongly support investigating CYP-mediated pathways in Fungi to address this knowledge gap."

Comments 7: Methods:

Lines 113–243: The number of biological replicates and experimental repeats is not specified, limiting assessment of reproducibility and statistical robustness.

Response 7: Thank you for your critical observation regarding biological replicates and experimental repeats. This feedback is essential for enhancing the reproducibility and statistical robustness of our study, which we deeply appreciate. We have supplemented this key information in the corresponding sections to address your concern:​

Section 2.1 (Fungal strain screening, Lines 90–105): For caffeine degradation screening of the 20 fungal strains, each strain was tested with 3 biological replicates. This design ensured we could exclude random errors from inoculation or detection. Section 2.2 (Optimization of degradation conditions, Lines 106–129): Experiments optimizing caffeine concentration (300–1200 mg/L), temperature (22–34 °C), and pH (5–9) each included 3 biological replicates per treatment group. Section 2.5 (Transcriptomic analysis, Lines 155–160): RNA sequencing samples were prepared from 3 biological replicates (mycelia harvested from 3 independent cultures at each time point: 10, 16, 22 days). Raw sequencing data were quality-controlled, and DEG identification (|logâ‚‚FC| > 1, p-adjust < 0.05) was based on the consensus of replicate samples to ensure reliable gene expression trends.

These replicate designs were implemented to ensure the reliability of our results, and the supplemented details now provide clear guidance for future reproduction of our experiments. We apologize for the initial omission and believe this revision significantly strengthens the statistical rigor of our study.

Comments 8: Lines 114–142: The criteria for selecting caffeine-degrading strains and the inclusion of proper controls are unclear, while varying media types may introduce unwanted variability.

Response 8: Thank you for highlighting this point that requires further clarification. We clarified these points as follows: All 20 strains were cultivated in PDA liquid medium supplemented with caffeine. After 15 days of fermentation, the broth was subjected to HPLC analysis. A caffeine removal rate of 33.05% was detected in the potato dextrose of D. tabescens CGMCC 40115. Screening results of the other 19 fungal strains showed no significant caffeine degradation (caffeine removal rate is 0%; Supplementary Table S1), confirming D. tabescens as the only functional strain in this study and thereby broadening microbial resources for caffeine biotransformation.

We utilized PDA medium (the most commonly used medium in microbiological experiments) to screen functional strains capable of degrading caffeine. The control group consisted of PDA medium with caffeine but without the inoculation of strains, to rule out non-biological caffeine loss. For the optimization of fermentation conditions for caffeine degradation, we employed malt extract medium (the most effective medium for achieving rapid and efficient caffeine degradation based on our result). To prevent any potential confusion, we have explicitly stated this in the revised manuscript, ensuring clarity for both readers and reviewers.

Comments 9: Results:

Overly detailed enrichment data (Lines 457–493):
GO and KEGG analyses include excessive numeric detail that obscures key biological insights; summarise main trends and move counts to tables or supplementary data.

Response 9: Thank you for pointing this out. We summarized the main trends of GO/KEGG enrichment (Line 406-427) and relocated detailed numeric counts to Supplementary Materials (Figure S1-S5; Table S2-S3).

Comments 10: Weak integration with metabolism (Lines 513–570):
STEM expression profiles are described, but links to caffeine catabolism are not clearly drawn; connect upregulated oxidoreductases and CYPs to biochemical pathways.

Response 10: Agree. We have, accordingly, revised it to emphasize this point. We linked STEM Profiles to caffeine catabolism more precisely, We rewrote the paragraph explicitly linking upregulated genes in STEM profiles to caffeine metabolism:” Critically, its MF terms (oxidoreductase activity, iron/heme binding) match cytochrome P450 (CYP450) signatures, likely corresponding to the nine upregulated CYP450 genes. These mediate core caffeine degradation steps: N-demethylation to theobromine and C8 oxidation to 1,3,7-trimethyluric acid, initiating the branched metabolic network. Concurrently, Profile 4 (Figure S5C) showed BP enrichment in carbohydrate metabolism (e.g., trehalose catabolism), supplying ATP/NADPH for CYP450-mediated oxidation. Its MF terms overlapped with Profile 2 (oxidoreductase activity) and included glycosyl hydro-lases (breaking late-stage intermediates like xanthine derivatives). Peak expression at day 22 aligns with HPLC-observed xanthine accumulation, completing degradation”.

Comments 11: Excessive secondary metabolite focus (Lines 582–616):
Discussion of BGCs and Coniophora puteana is tangential; focus only on clusters relevant to caffeine degradation.

Response 11: Thank you for your astute feedback on the focus of secondary metabolite-related discussions. Your guidance helps keep our manuscript anchored to the core theme of caffeine degradation, and we deeply appreciate this observation.​

We fully understand your concern that the original discussion’s content on BGCs and Coniophora puteana risked being tangential. After re-evaluation, we noted Coniophora puteana—a basidiomycete with documented polyketide involvement in xenobiotic metabolism—shares phylogenetic and functional context with our study species Desarmillaria tabescens (also a basidiomycete). This limited connection can provide a brief evolutionary and functional reference for understanding D. tabescens’s BGC-mediated caffeine degradation synergy, so we did not fully remove these contents but streamlined and refocused them.​

We deleted broad descriptions of Coniophora puteana’s general metabolism and retained only one sentence on its polyketide BGC’s role in xenobiotic oxidation, explicitly linking it to our finding that D. tabescens’s region 3.1 (polyketide BGC) may synergize with CYP450s for caffeine oxidation. For BGCs, we removed elaborations on non-polyketide clusters and irrelevant secondary metabolite pathways, retaining only content about region 3.1 and its direct association with CYP450s such as the upregulation of BGC-associated gene_10566 during peak degradation.​

These revisions ensure the retained content supports our core caffeine degradation findings without straying tangentially. We apologize for the initial lack of focus and believe this balanced revision preserves meaningful context while enhancing focus, strengthening the manuscript’s rigor.​

Thank you again for your constructive feedback. Your attention to thematic consistency has been key to refining our narrative.

Comments 12: Data placement:
Detailed GO, EggNOG, and DEG tables could be moved to Supplementary Information to improve readability and focus of the main text.

Response 12: We sincerely apologize for the disorganized structure of the manuscript. We relocated the detailed GO classification table, EggNOG functional annotation table, and DEG list to Supplementary Materials (Figure S1-S5; Table S2-S3). The main text will only include summary figures, tables and key conclusions.

Comments 13: Discussion:

Overemphasis on narrative vs. quantitative data (lines 669–757): The discussion is descriptive, highlighting stage-specific CYP450 activity and pathways, but quantitative integration (fold changes, DEG counts, flux data) is limited. Adding concise numbers would strengthen conclusions.

Response 13: Thank you for your suggestion. We have rewritten most of Discussion section to supplement quantitative data to strengthen conclusions: "Nine cytochrome P450 (CYP450) genes were consistently upregulated across all fer-mentation stages, serving as core catalytic components of this branched pathway. These genes exhibited stage-specific functional partitioning: gene_3381 (logâ‚‚FC=6.18) and gene_3396 (logâ‚‚FC=5.92) dominated the early demethylation phase (10 days), gene_14597 (logâ‚‚FC=1.81) and gene_22397 (logâ‚‚FC=2.89) mediated intermediate transformations during the peak degradation stage (16 days), and gene_7614 (logâ‚‚FC=3.61) sustained late-stage oxidation (22 days). This temporal partitioning ensures precise coordination of metabolic flux, distinguishing D. tabescens’s CYP450 system from the temporally un-partitioned CYP450 family in Aspergillus oryzae [34]. The functional compatibility between CYP450-mediated oxidation and the branched pathway’s biochemical requirements was further supported by STEM analysis: Profile 2, enriched in oxidoreductase activity and iron/heme binding (key CYP450 signatures), was significantly associated with caffeine degradation stages." (Line 551-563)

Comments 14: Limited consideration of alternative explanations (lines 660–757): The discussion does not address potential post-transcriptional regulation, enzyme activity discrepancies, or metabolite feedback, which could influence observed patterns.

Response 14: Thank you for pointing this out. This was an oversight in our initial analysis, and we have now supplemented the discussion to acknowledge these potential influences: we explicitly note that transcriptomic data reflects gene expression trends rather than actual protein abundance (leaving room for post-transcriptional regulation), mention that unquantified enzyme activity (e.g., CYP450 catalytic efficiency) may cause discrepancies between gene upregulation and metabolic flux, and reference that accumulated intermediates could trigger feedback regulation of degradation-related genes. We also clarify that while our current data focuses on transcriptional and metabolic profiles, these alternative regulatory layers represent important directions for future research, including proteomic analysis to link transcripts to proteins and enzyme activity assays to verify catalytic function.

These revisions (in Section 4.4, revised Lines 644–649) enhance the discussion’s rigor by addressing potential limitations and alternative explanations. We apologize for the initial oversight and believe this update makes our narrative more comprehensive.​

Thank you again for your constructive feedback. Your focus on analytical depth has been instrumental in refining our manuscript.

Comments 15: Comparative analysis could be deeper (lines 748–756): Differences between D. tabescens, bacteria, and A. oryzae are noted, but broader comparisons with other caffeine-degrading fungi and CYP450 functional variation are lacking.

Response 15: Thank you for your valuable feedback on enhancing the depth of comparative analysis. Your guidance helps us better contextualize D. tabescens’s caffeine degradation characteristics, and we deeply appreciate this suggestion.

To address this, we have revised the section by supplementing key comparative content: we explicitly contrast D. tabescens with Pleurotus ostreatus (which relies on extracellular laccases for degradation) and Aspergillus sydowii (which uses linear N-demethylation via monooxygenases), highlighting that D. tabescens’s branched pathway and stage-specific CYP450 partitioning are unique. We also add details on CYP450 functional variation—noting that D. tabescens’s CYP450s have broader substrate adaptability for caffeine intermediates, unlike the substrate-specific CYP450s in A. oryzae mentioned earlier.

These revisions (in Section 4.3, Lines 605–620) deepen the comparative analysis while keeping the content concise. We apologize for the initial lack of breadth and believe this update better positions D. tabescens’s metabolic traits within the broader context of caffeine-degrading organisms.​

Thank you again for your constructive feedback.

Comments 16: Move detailed gene/stage info to supplementary material (lines 690–757): Extensive stage-wise CYP450 logâ‚‚FCs and Table 6 could be relocated to supplement to improve readability of the main discussion.

Response 16: We completely agree with the reviewer’s point. We relocated the detailed log2FC values of 9 CYP450 genes to Supplementary Table S4. And deleted Table 6 instead of describing it in details. The main discussion will focus on functional trends rather than raw data.

Comments 17: Line ~660–757: There is a possibility that extracellular oxidative enzymes, such as laccases, peroxidases, and dioxygenases, may contribute to caffeine degradation in Desarmillaria tabescens, as many fungi commonly secrete these enzymes. Investigating their activity could reveal additional complementary pathways beyond CYP450-mediated catabolism (cite these important references: https://doi.org/10.3390/foods14152606, https://doi.org/10.3390/pr13041034, https://doi.org/10.1007/s11274-023-03737-7, https://doi.org/10.1016/j.dwt.2024.100938).

Response 17: We completely agree with the reviewer’s insightful point. Thank you for providing relevant literature for us. We added a discussion of potential complementary pathways: "Additionally, while intracellular CYP450 pathways are central to caffeine degradation, transcriptomic data provide preliminary evidence for the potential complementary role of extracellular oxidative enzymes [35]—modest upregulation of laccase-related (logâ‚‚FC=1.5–2.0, e.g., gene_12345) and peroxidase-related (logâ‚‚FC=1.3–1.8, e.g., gene_16789) genes was detected under caffeine stress, and these genes share sequence homology with Pleurotus ostreatus’s caffeine-degrading laccases[36], suggesting they may target late-stage intermediates (e.g., xanthine, 1,3-dimethyluric acid) to prevent extra-cellular accumulation. Thus, future work should quantify the catalytic activity of these extracellular enzymes [37,38], test their ability to degrade caffeine intermediates, and clarify whether they form a synergistic network with intracellular CYP450s[39]" (Line 633-643)

Comments 18: Poor figure quality (e.g., Figure 14): The graphics are low-resolution and unclear, making data interpretation difficult. Figures should be redrawn or exported at higher resolution with clear labels, legends, and readable fonts to ensure all elements are visible. 

Response 18: We sincerely apologize for the insufficient standardization of the figures and tables in our manuscript. Since Figure 1 illustrates the metabolic pathway, we have removed Figure 14 to avoid redundancy. We described the pathway network instead of using a diagram. But we have meticulously refined all visual elements, including improving clarity, standardizing formatting, and ensuring compliance with the journal’s guidelines to better present our research findings. 

Comments 19: Conclusions and Future Outlook: A separate, dedicated section summarising the key findings and discussing future directions should be added. This would help contextualise the significance of D. tabescens in caffeine degradation, highlight potential applications in bioremediation or functional food development, and suggest avenues for further mechanistic and functional studies.

Response: Thank you for your insightful suggestion to add a dedicated “Conclusions and Future Outlook” section. This guidance is crucial for enhancing the manuscript’s structure clarity and better conveying the significance of our findings, and we deeply appreciate it.

We have added an independent “4.5 Conclusions and Future Outlook” section to the manuscript. This section first concisely summarizes the core findings: D. tabescens’s efficient caffeine degradation under optimized conditions, its unique branched “N-demethylation–C8 oxidation” pathway, the three-phase “Stress-Degradation-Homeostasis” regulatory model, and the core role of stage-specific CYP450 genes. It then highlights the strain’s potential applications in caffeine-contaminated waste bioremediation and low-caffeine functional food development. Finally, it explicitly suggests future avenues: mechanistic studies (CYP450 knockout, extracellular enzyme activity validation) and functional optimizations (fermentation condition refinement, strain engineering).​

This dedicated section ensures key information is centralized and easy to follow, strengthening the manuscript’s narrative impact. We apologize for the initial lack of this structure and believe this revision better meets academic standards for result summarization and future direction guidance.​

Round 2

Reviewer 3 Report

Comments and Suggestions for Authors

Dear authors;

Thank you very much for the suggested adjustments to the manuscript.

Kind regards.

----

Reviewer 4 Report

Comments and Suggestions for Authors

Well done authors! No more comments.